# U-Pb Ages, O Isotope Compositions, Raman Spectrum, and Geochemistry of Cassiterites from the Xi'ao Copper-Tin Polymetallic Deposit in Gejiu District, Yunnan Province

**Yuehua Zhao [1], Shouyu Chen [1,2,\*], Yuqiang Huang [1], Jiangnan Zhao [1,2], Xiang Tong [3] and Xingshou Chen [3]**

[1] Faculty of Earth Resources, China University of Geosciences, Wuhan 430074, China; yuehuazhao@outlook.com (Y.Z.); hyq_person@163.com (Y.H.); zjn198402@163.com (J.Z.)
[2] State Key Laboratory of Geological Processes and Mineral Resources, China University of Geosciences, Wuhan 430074, China
[3] Yunnan Tin Company Group, Gejiu 610000, China; TXYXJT123@163.com (X.T.); chenxs123@126.com (X.C.)
[\*] Correspondence: sychen@cug.edu.cn

**Abstract:** The Xi'ao Cu-Sn polymetallic deposit is located in the inner alteration zone of the Laoka granite. The ore bodies extend to 400 m in the granite rock and primarily occur with fluorite and potassic alterations. Two cassiterite samples of altered rock-type ore and one tourmaline vein-type ore in the Xi'ao Cu-Sn polymetallic deposit yielded U-Pb ages of 83.3 ± 2.1 Ma, 84.9 ± 1.7 Ma, and 84.0 ± 5.6 Ma, respectively. The Raman spectrum peak values of $A_{1g}$ were shifted to a lower frequency, possibly due to the substitution of Sn with Nb, Ta, Fe, and Mn. Measured $\delta^{18}O$ values of cassiterite samples and calculated $\delta^{18}O_{H2O}$ values for the ore-forming fluid indicate that the latter was mostly derived from magma. The high Fe and Mn abundances for cassiterite are consistent with those of hydrothermal origin. The Nb, Ta, and Ti contents indicate that cassiterites in the Xi'ao deposit likely formed in a metallogenic environment that was largely affected by granitic magmatism. Therefore, we conclude that the Xi'ao deposit is a magmatic hydrothermal deposit.

**Keywords:** cassiterite; U-Pb ages; O isotope; Raman spectrum; Xi'ao Cu-Sn deposit; Gejiu district

## 1. Introduction

Cassiterite is one of the main tin minerals that is widespread in tin deposits; it can precipitate from hydrothermal ore-forming fluids over a very broad range of P-T-X conditions, and it is resistant to metamorphism, hydrothermal alteration, weathering, and abrasion [1,2]. Hence, it is likely that cassiterite effectively preserves primary information about the trace and rare-earth element geochemistry of the ore-forming fluid, which can provide important information for studies on the genesis of ore deposits [1–7]. Cassiterite belongs to the rutile group ($M^{4+}O_2$). It generally has high U and low Pb contents in its crystal structure and a high Pb closure temperature (560–860 °C) [8]. In recent years, great progress has been made in the use of cassiterite to determine the metallogenic ages of Sn polymetallic deposits [7,9–17]. Yuan et al. [18] obtained in situ LA-ICP-MS U-Pb age data (159.9 ± 1.9 Ma) and ID-TIMS U-Pb age data (158.2 ± 0.4 Ma) from cassiterite in the Furong Sn polymetallic deposit. Zhang et al. [16] provided further evidence to constrain the timing of granitic magmatism and hydrothermal mineralization by using LA-MC-ICP-MS U-Pb dating to calculate ages for igneous zircon and hydrothermal cassiterite. Compared with the metallogenic ages obtained from altered

minerals, ages obtained directly from cassiterite can more accurately reflect the timing of ore deposit formation [7].

China is extremely rich in tin resources, possessing 32% of the world's total Sn resources [19]. The Gejiu tin polymetallic district in Yunnan is well known for its large tin reserves containing approximately 335.74 Mt of Sn ores, 357.11 Mt of Cu ores, and 400 Mt of Pb-Zn ores. In the past few decades, this tin district has been extensively studied. Dating is a powerful tool used to determine mineralization characteristics and understand the ore genesis process [9,20–22]. Attempts to date the mineralization of the Gejiu district have been primarily performed on hydrothermal minerals ([40]Ar-[39]Ar ages of micas, Re-Os ages of molybdenites) [23–26], but in some cases, the ages of these minerals may be inconsistent or inaccurate (the [40]Ar-[39]Ar and K-Ar ages range from 43.49 ± 0.87 Ma to 87.5 ± 0.6 Ma) [24,25,27,28]. Thus, ages of ore minerals are still needed to provide precise constraints on the timing of mineralization processes.

In Gejiu district, the Xi'ao ore field, an altered granite-type Cu-Sn polymetallic deposit (containing > 10 Mt Sn + Cu [29]), has been discovered in the inner alteration zone of the western edge of the Laochang–Kafang granite. In this deposit, the granite–marble contact zones rarely contain skarn minerals, and the intensity and scale of the mineralization are substantially constrained by the wall rock alteration and spatial distribution of granite [29,30]. The genesis of this deposit remains unclear, and additional studies are needed. Currently, most studies of cassiterite in the Gejiu district have focused on its mineral typomorphic characteristics [31–34], and some studies have investigated the geochronology and geochemistry of cassiterites from the Gaosong and Laochang deposits (two ore fields in Gejiu district) [14,35,36]. However, few studies have been performed on cassiterites from Xi'ao deposit, especially for altered rock-type cassiterites.

In this paper, cassiterite samples were collected from altered rock-type and tourmaline vein-type ores in the Xi'ao Cu-Sn polymetallic deposit and studied to constrain the timing of tin mineralization and precipitation environment of cassiterite. Consequently, the mineralogy of cassiterites was investigated by cathodoluminescence (CL) imagery and Raman spectroscopy. The geochemical signatures and oxygen isotopes of these minerals were also analyzed by using electron microprobe analyzer (EPMA) and isotope ratio mass spectrometry (MAT 253). In addition, the U-Pb ages of cassiterites were also measured by LA-ICP-MS. These results provide further insights into ore genesis in the Xi'ao deposit and highlight the potential of using cassiterite as a monitor of hydrothermal processes.

## 2. Geological Setting

The Gejiu mining district is located in the southeastern region of Yunnan Province. Tectonically, the Gejiu tin polymetallic deposit is located at the junction of the Yangtze Craton and Cathaysia and Indochina blocks (Figure 1). This deposit represents an important part of the southeast Yunnan tin polymetallic belt [37,38]. Sedimentary sequences in Gejiu district comprise Cambrian to Quaternary rocks, but Cretaceous rocks are lacking because of episodic uplift/erosion associated with Indosinian and Yanshanian tectonic events. Most outcrops in the Gejiu area comprise carbonate rocks of the Triassic Gejiu Formation and fine-grained clastic sediments and carbonates of the Falang Formation, the former of which are the main ore-hosting rocks. Numerous faults are present in the area, including the North–Northeast- (NNE) trending Longchahe, Jiaodingshan, and Yangjiatian faults, as well as the Northwest- (NW) trending Baishachong fault and the North–South- (NS) trending Gejiu fault. The Gejiu fault divides the study area into its eastern and western sectors. The eastern area, which contains 90% of the tin reserves in the Gejiu district [38], is dominated by the Wuzhishan anticlinorium, which includes five ore deposit areas; from north to south, these areas are: Malage, Songshujiao, Gaosong, Laochang, and Kafang (Figure 1).

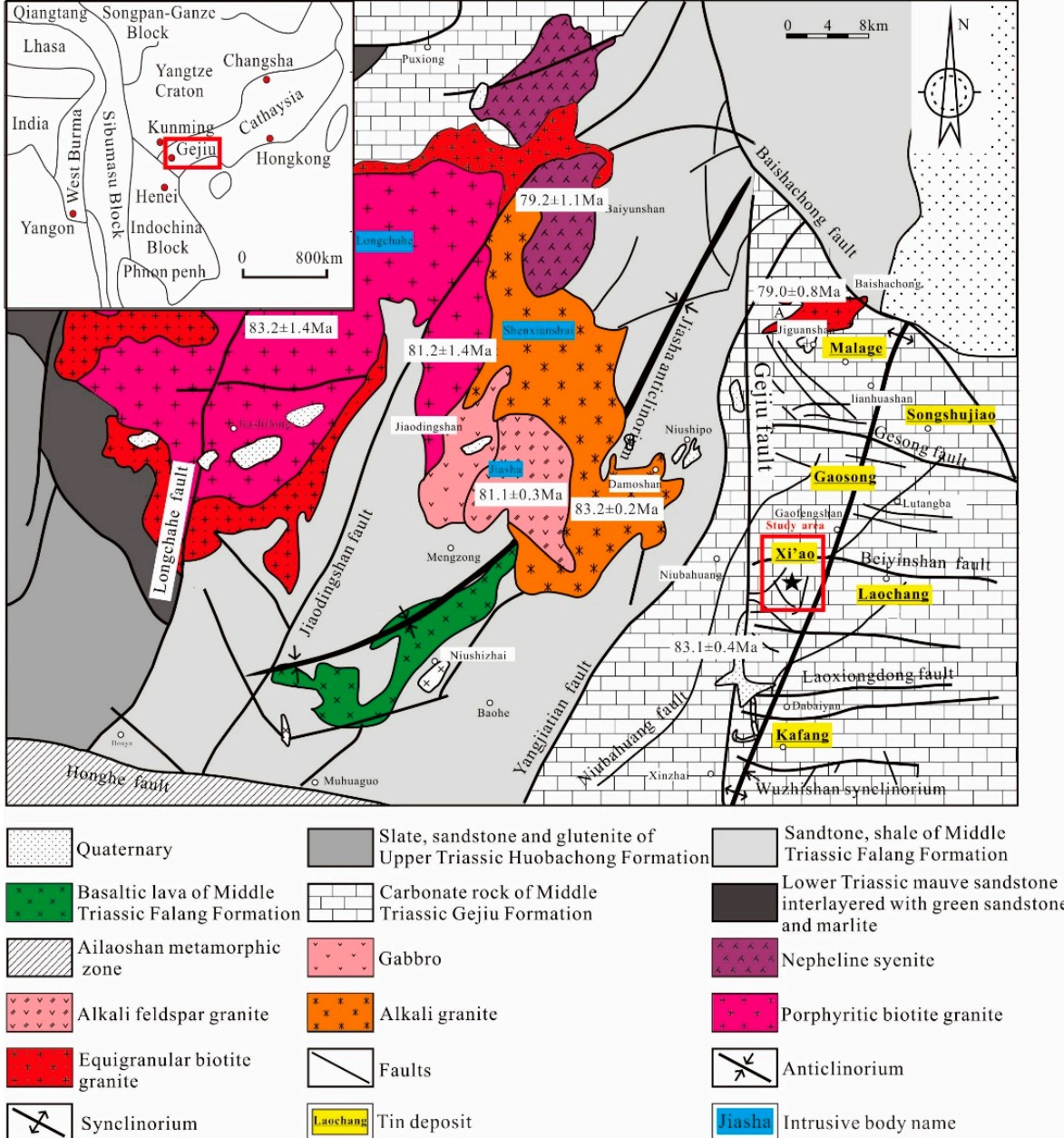

**Figure 1.** Geotectonic location (after [39]) and geological sketch map of Gejiu area, Yunnan Province (modified after [40]). The star shows the location of the Xi'ao Cu-Sn polymetallic deposit.

Frequent multistage magmatic activity occurred in the Gejiu area. Gejiu magmatic intrusions (emplacement ages are marked in Figure 1), which show a large variability, including gabbro, nepheline syenite, alkali feldspar granite, alkali granite, porphyritic biotite granite, and granular biotite granite, are widely exposed in the western area of Geiju but are sparsely distributed in the eastern area. The ages of granites in Gejiu range from 53 Ma to 147 Ma according to dating obtained by different methods such as Ar-Ar/K-Ar, Rb-Sr, LA-ICP-MS/TIMS/SHRIMP (Sensitive High-Resolution Ion Microprobe) U–Pb, and Re–Os dating [24,25,41–45]. The latest chronological studies have shown that the magmatism in Gejiu district occurred during the late Cretaceous period (zircon U-Pb ages vary from 76.6 ± 3.6 Ma to 85.0 ± 0.85 Ma, Table 1). Trace elements and Sr-Nd-Hf isotopes demonstrate that the Gejiu granites were mainly derived from melting of continental crust in an intraplate environment [43]. The predominant granitoids and the synchronous mafic and ultramafic rocks in Gejiu thus constitute a bimodal magmatic association controlled by lithospheric extension and asthenosphere upwelling within the western Cathaysia block in Late Cretaceous [46–49], indicating a late Yanshanian intracontinental extensional tectonic context [40,43,50,51]. Previous

studies have also shown that the granites emplaced in Yanshanian are closely temporally and spatially related to the Gejiu tin polymetallic deposit [23,40,43].

**Table 1.** The ages of magmatic intrusions in Gejiu district.

| Pluton Name | Lithology | Research Object | Dating Method | Age | Petrogenesis | Reference |
|---|---|---|---|---|---|---|
| Longchahe | Porphyritic biotite granite | biotite | K-Ar | 100~115 Ma | A-type; peraluminous, alkaline granite | [51] |
| | | whole rock | Rb-Sr | 147 ± 3 Ma | | |
| | | zircon | U-Pb | 82.0 ± 0.3 Ma~83.2 ± 1.4 Ma | | [43] |
| Masong | Porphyritic biotite granite | biotite | K-Ar | 100~102 Ma | A-type and S-type; metaluminous – peraluminous, calc-alkaline granite | [51] |
| | | K-feldspar | K-Ar | 91.5~116 Ma | | |
| | | whole rock | Rb-Sr | 90.4 ± 6.3Ma | | |
| | | zircon | U-Pb | 82.8 ± 1.7 Ma | | [43] |
| Shenxianshui | Equigranular granite | biotite | K-Ar | 72~87 Ma | A-type; peraluminous, alkaline granite | [51] |
| | | whole rock | Rb-Sr | 84.4 ± 1.1 Ma | | |
| | | zircon | U-Pb | 81.0 ± 0.52 Ma ~81.4 ± 0.4 Ma | | [43] |
| Baishachong | Equigranular granite | biotite | K-Ar | 53 Ma | Peraluminous, calc-alkaline granite | [52] |
| | | whole rock | Rb-Sr | 81.0 ± 2 Ma | | |
| | | zircon | U-Pb | 77.4 ± 2.5 Ma | | [43] |
| Laoka | Equigranular granite | biotite | K-Ar | 64 Ma ~80 Ma | S-type; peraluminous, calc-alkaline | [52] |
| | | whole rock | Rb-Sr | 81.0 ± 4.9 Ma | | |
| | | zircon | U-Pb | 85.0 ± 0.85 Ma | | [41] |
| | Porphyritic granite | zircon | U-Pb | 83.3 ± 1.6 Ma | | [43] |
| Baiyunshan | Alkali feldspar granite | biotite | K-Ar | 59.5 Ma~62 Ma | / | [52] |
| | | whole rock | Rb-Sr | 94.3 ± 2.4 Ma | | |
| | | zircon | U-Pb | 76.6 ± 3.6 Ma | | [53] |
| Jiasha | Gabbro | zircon | U-Pb | 84.0 ± 0.6 Ma | / | [54] |
| Lamprophyre | | zircon | U-Pb | 77.2 ± 2.4 Ma | / | [53] |

## 3. Ore Geology of the Xi'ao Deposit

The Xi'ao Cu-Sn polymetallic deposit is located in the western part of the Laochang ore field (Figure 1), and the mining area is approximately 25 km² [55]. The outcrop of the area mainly comprises Triassic Gejiu Formation carbonate rocks, which are the main ore-hosting rocks. The NW-trending Huangmaoshan and East–West- (EW) trending Wanzijie anticlines, which are the subsidiary anticlines of the Wuzhishan anticlinorium, are the main fold structures. The faults comprise three groups—NW-, NE-, and NS-trending—all of which are closely associated with mineralizations.

The Laoka equigranular biotite granite intruded into the Gejiu Formation in the Late Cretaceous (85.0 ± 0.85 Ma, [43]) at depths of 200–1800 m beneath the surface. It is mainly distributed in the Laochang and Kafang ore fields. Rock-forming minerals in equigranular biotite granite include K-feldspar (38%), plagioclase (25.2%), quartz (32.8%), and biotite (4%). The accessory minerals are zircon, apatite, titanite, allanite, monazite, tourmaline, and fluorite. Petrographic and geochemical features suggested that the hidden Laoka biotite granite has affinities of calc-alkaline S-type granite with higher contents of Sn (33.3 ppm), Cu (12.8 ppm), W (5.3 ppm), and mineralization-associated elements (F: 2500 ppm and Cl: 250 ppm) than normal granite [56,57]. It is peraluminous and enriched in silica and potassium (SiO$_2$ content is high, SiO$_2$ > 74%; K and Na rich, K$_2$O + NaO = 7.19%, K$_2$O/Na$_2$O = 6.8; aluminum saturation, Al/(K$_2$O + Na$_2$O) = 1.83; low oxidation rate, O$x$ = FeO/(FeO + Fe$_2$O$_3$) = 0.46 and highly fractionated, DI = 85~95) [29,38]. In addition, this granite is highly evolved and fractionated and possibly formed during the late evolution stage of the Gejiu granite [43,51].

The ore bodies are located in the inner alteration zone along the western edge of the Laoka granite (Figure 2) approximately 1000 m below the surface [29]. These ore bodies are mainly controlled by the EW- trending faults, occur as veins and lenticular bodies, and extend to 400 m into the granite. The grades of tin and copper range from 0.2% to 1.34% and from 0.3% to 3.0%, with metal contents of 127.6 and 69.8 thousand tons, respectively [50]. The alteration of rocks is intensive, and

skarn is poorly developed in the ore field. The country rock alterations are potassic alteration, tourmalinization, fluoritization, pyritization, silicification, chloritization, and carbonatization. The alteration zone boundary is vague. The alteration zones are classified into two major types—potassic and epidote-chlorite. The former is closely related to mineralization [56,58,59]. The mineral assemblages are listed in Table 2.

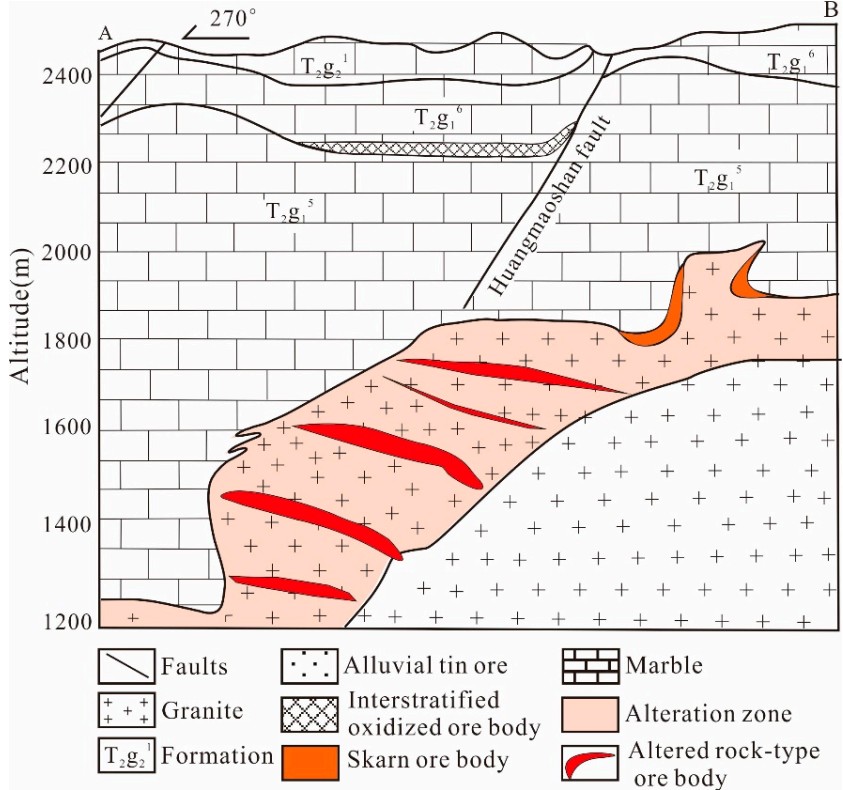

**Figure 2.** Mineral geological map of 164 exploration lines (after [29]).

**Table 2.** The mineral assemblage characteristics of altered-rock type ores in Xi'ao deposit.

| Mineral Assemblage Type | Minerals | Ore Structure | Mineralization Significance |
|---|---|---|---|
| Potassium-fluoride-sulfide | K-feldspar, pyrite, chalcopyrite, and less fluorite, quartz, tourmaline, mica, etc. | Block, crumb, veinlet and disseminated | Big |
| Epidote and chlorite-pyritization-fluoride | Quartz, plagioclase, k-feldspar, biotite, epidote, chlorite, pyrite, fluorite, tourmaline, etc. | Veinlet and disseminated. | Small |

Based on the occurrence, locality, and paragenesis of tin ore, two mineralization types were collected from the mining area, i.e., altered rock- and tourmaline vein-type ore (Figure 3).

Altered rock-type Sn polymetallic ore (Figure 3a,b) (due to the paucity of studies, we name it provisionally here) represents a newly discovered mineralization type, providing an interesting direction for deep prospecting in the future. The distribution of orebodies is controlled by EW-trending fractures, which are located in the inner alteration zone of the Laoka granite edge (approximately 10 m to 400 m within the rock mass). The main alteration types associated with mineralization are potassic alteration and chloritization. In the inner granite alteration zone, ore bodies are found only in K-altered granite. Orebodies mostly occur as veins, veinlets, or banded bodies such as parallel arrangements. When intersecting fractures, lenticular ores are also present. The average grade of Sn is 0.51 wt %, and it can reach up to 10 wt %. In potassic alteration zone, the original plagioclase, quartz, and other minerals are overprinted and essentially replaced by secondary K-feldspar, perthite, and microcline. The main metallic minerals are chalcopyrite, pyrrhotite, sphalerite, cassiterite, pyrite, scheelite, and wolframite (Figure 3d,g). These minerals are

mainly disseminated (rich), and stringers fill the structures (barren) of altered granite (Figure 3d,e). The gangue minerals mainly contain K-feldspar, fluorite, and tourmaline, and less mica, quartz, epidote, chlorite, and apatite. Ore textures are mainly metasomatic relict, poikilitic, and skeletal (Figure 3d,e).

Tourmaline vein-type ores (Figure 3c) occur as veins, veinlets, and stockwork and are predominantly controlled by fractures. The vein ores are present at a wide range of depths, from proximal to the granite to near the ground surface. The ore minerals, which are mainly disseminated, mainly contain chalcopyrite, pyrite, cassiterite, and arsenopyrite with lesser sphalerite and pyrrhotite (Figure 3h). The gangue minerals mainly include tourmaline and fluorite, with little quartz (Figure 3i).

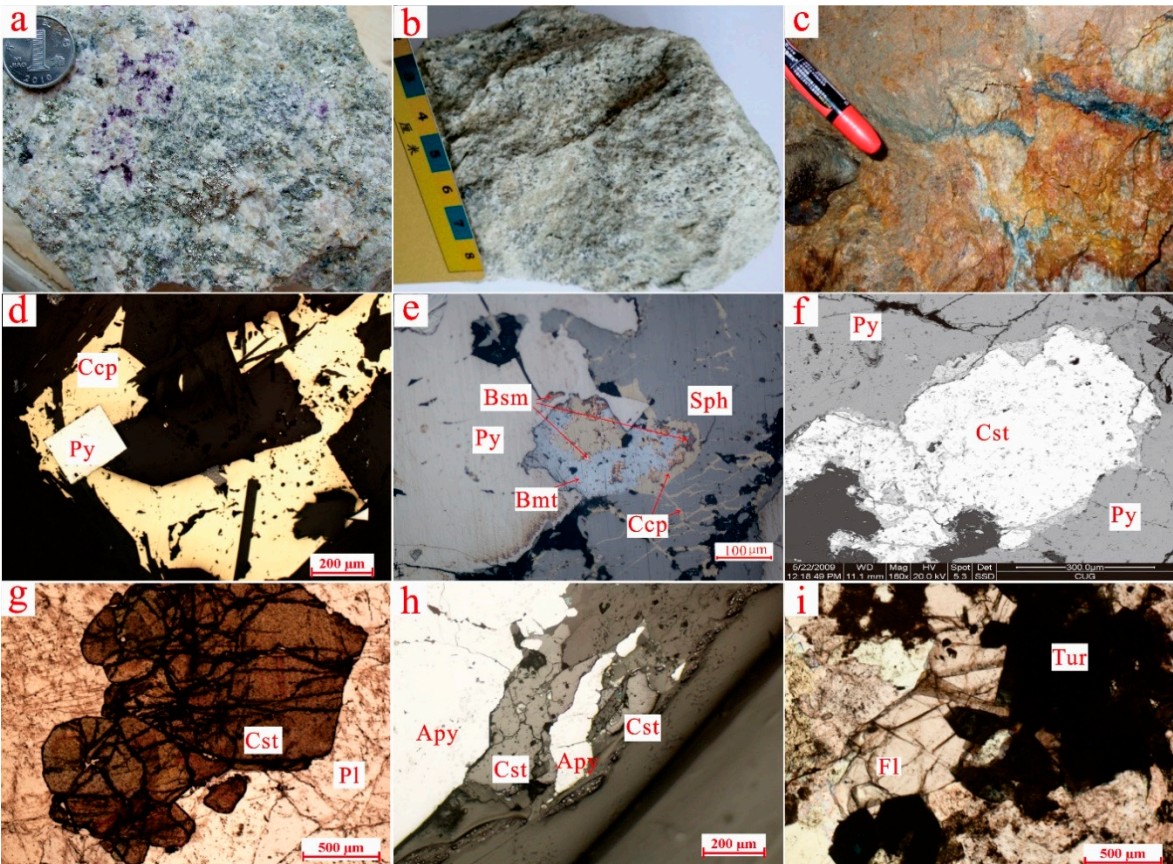

**Figure 3.** Photographs of tin ores from the Xi'ao Cu-Sn deposit; (**a**) is altered rock-type ore (XA-3) and mainly contains chalcopyrite, pyrite, sphalerite, quartz, fluorite, and cassiterite; (**d**) and (**e**) are photomicrographs of XA-3; (**f**) is the BSE image of XA-3; (**b**) is altered rock-type ore (XA-5) and mainly contains cassiterite, feldspar, and quartz; (**g**) is the photomicrographs of XA-5; (**c**) is tourmaline vein-type ore (XA-4) and mainly contains tourmaline, cassiterite, arsenopyrite, and fluorite; (h) and (**i**) is the photomicrographs of XA-4. Abbreviations: Ccp, chalcopyrite; Py, pyrite; Sph, sphalerite; Bsm, Bismuth; Bmt, bismuthinite; Cst, cassiterite; Pl, feldspar; Apy, arsenopyrite; Fl, fluorite; Tur, tourmaline.

## 4. Samples and Analytical Methods

Samples XA-3 (Figure 3a) and XA-5 (Figure 3b) were obtained from altered rock-type ore and tourmaline vein-type ore in the Xi'ao 1800 m adit, respectively. XA-4 (Figure 3c) was obtained from altered rock-type ore in granite at Fengliushan. Prior to analysis, samples were crushed, and cassiterite grains were separated using heavy liquid and magnetic separation techniques. They were then handpicked under a binocular microscope, mounted in an epoxy resin disc, and polished.

Cassiterite samples were examined carefully under the binocular microscope and scanning electron microscope to observe mineral and fluid inclusions and cracks. We chose the clean surface of cassiterites when experimenting in order to avoid fractures and inclusions, as common Pb is mostly associated with fluid inclusions [10].

Cathodoluminescence (CL) images of three cassiterite samples were obtained at Wuhan SampleSolution Analytical Technology Co., Ltd., Wuhan, China, using an analytical scanning electron microscope (JSM-IT100) (Japan Electron Optics Laboratory Co., Ltd, Tokyo, Japan) connected to a GATAN MINICL system (Gatan, Inc., Pleasanton, USA). The operating conditions included an accelerating potential of 10 kV, a temperature of 20 °C, and an image acquisition time of 35 s/sheet. To compare the CL images of cassiterite grains, specific analytical conditions were adopted during the experiment such that the instrument automatically adjusted the parameters to obtain an image with the highest resolution after the scale was changed.

Raman spectra of three cassiterite samples were determined at the State Key Laboratory of Geological Processes and Mineral Resources (GPMA), China University of Geosciences (Wuhan), using a Renishaw RW-1000 Raman microspectrometer (Renishaw, London, United Kingdom). An argon ion laser with a wavelength of 514.5 nm was used for detection at 25 °C. The laser beam was focused on a small area measuring 2.0 μm. Working conditions: the laser power was 0.2 W, the collected time of Raman spectrum was 60 s, and the spectral resolution was ±1 nm$^{-1}$. The spectrograms were processed using OriginPro 8.0 (OriginLab, Northampton, USA).

Mineral compositions were determined at the State Key Laboratory of Geological Processes and Mineral Resources, China University of Geosciences (Wuhan) using a JEOL JXA-8100 Electron Probe Micro Analyzer equipped with four wavelength-dispersive spectrometers (WDS) (Japan Electron Optics Laboratory Co., Ltd, Tokyo, Japan). The samples were first coated with a thin conductive carbon film prior to their analysis. The precautions suggested by Zhang and Yang [60] were used to minimize differences in the thicknesses of carbon films between samples and to obtain a uniform coating of approximately 20 nm. During the analysis of these minerals, an accelerating voltage of 15 kV, a beam current of 20 nA, and a spot size of 2 μm were used. Data were corrected online using a modified ZAF (atomic number, absorption, fluorescence) correction procedure. The peak counting time was 10 s for Sn, W, Ta, Si, Fe, and Nb, and 20 s for In, Ti, and Mn. The background counting time was one-half of the peak counting time on the high- and low-energy background positions. The following standards were used: cassiterite (Sn), tungsten (W), tantalum (Ta), olivine (Si), indium phosphide (In), pyrope garnet (Fe), niobium (Nb), rhodonite (Mn), and rutile (Ti). Analytical uncertainty was estimated at <1 wt % error for major elements and up to 5 wt % for trace elements.

Pure cassiterite separates were prepared for oxygen isotope analysis. The O isotope compositions were analyzed following the BrF$_5$ method [61]. The δ$^{18}$O values of these separates were determined on a Finnigan MAT 253 ratio mass spectrometer (Thermo Fisher Scientific, USA). The O isotope results are reported relative to V-SMOW (Vienna standard mean ocean water), and the analytical precision is ±0.2‰ for δ$^{18}$O. The degree of isotopic fractionation between cassiterite and water was calculated using Equation (1) [62] at the minimum trapping temperature, which was defined based on the average homogenization temperatures of the fluid inclusions in each quartz and calcite sample, as reported by Liao et al. [50]:

$$\delta^{18}O_{water-SMOW} = \delta^{18}O_{V-SMOW} - 10^3 \ln\alpha_{cassiterite-water} = \delta^{18}O_{V-SMOW} - (10.13 \times 10^6/T^2 - 26.09 \times 10^3/T + 12.58) \quad (1)$$

In situ U-Pb cassiterite dating was performed using a Neptune ICP-MS (Thermo Fisher Scientific ICAP Q) (Thermo Fisher Scientific, USA) coupled with an ESI 193 nm COHERENT Compex Pro 102F Excimer laser ablation system (Cohernet Coherent Inc., Santa Clara, USA) at the State Key Laboratory of Geological Processes and Mineral Resources, China University of Geosciences (Wuhan). The Squid smoothing device was used to reduce statistical error induced by laser ablation pulses and to improve the data quality [63,64]. Helium gas carrying the aerosol of the ablated sample was mixed with argon carrier gas and nitrogen as an additional di-atomic gas to enhance sensitivity before finally flowing into the ICP. Typical gas flow settings for the Ar cooling gas, Ar auxiliary gas, and He carrier gas during the course of this study were 15 L/min, 0.75 L/min, and 0.86 L/min, respectively. The samples were analyzed using an energy density of 5 J/cm$^2$, a spot size of 32 μm, and a laser pulse frequency of 10 Hz. NIST SRM 610 (National Institute of Standards and Technology, Standard Reference Material 610) and an in-lab cassiterite standard AY-4 were used as external calibration standards. AY-4 was collected from the skarn orebody in the Anyuan tin deposit of the Furong orefield in the middle Nanling Range. This cassiterite sample has been well studied using ID-TI0MS, and it has a U-Pb age

of 158.2 ± 0.4 Ma [18]. Details can be found in Yuan et al. [13,18]. NIST SRM 610 was analyzed once after every ten analyses; AY-4 was analyzed twice after every five analyses. Each spot analysis incorporated approximately 20 s of background acquisition followed by 40 s of sample data acquisition. Isotopes were measured in time-resolved mode. For U-Pb dating, dwell times for each mass scan were 15 ms for [204]Pb, [206]Pb, [208]Pb, [238]U, and [235]U, and 25 ms for [207]Pb. Data errors of single spot were 1σ. Raw data reduction was performed off-line using ICPMSDataCal software (Liu Yongsheng, China University of Geosciences, China) [65,66]. The uncertainty of single populations, ratio uncertainty of the AY-4 reference material, and decay constant uncertainties were propagated to the ultimate results of the samples during the process of data reduction by ICPMSDataCal 10.1 [66]. Tera-Wasserburg concordia lower intercept age calculations were processed using Isoplot 3.0 (Kenneth R. Ludwig, United States Geological Survey, USA) [67].

## 5. Results

### 5.1. Color and CL Images

The typomorphic characteristics of the color of cassiterite have been well studied by researchers due to its easily recognized and distinguishable features [4,12,31–34,68].

As shown in Figure 4, the colors of the three cassiterite samples varied widely (e.g., black, dark brown, brown, red brown, light brown, gray, colorless). The colors of altered rock-type cassiterites varied from colorless to dark, while those of tourmaline vein-type cassiterites were mainly dark. In addition, the grain sizes of cassiterites that formed in the same ore-forming setting were uniform, while they varied widely among cassiterites of different genetic types, ranging from $n \times 10$ μm to $n \times$ 100 μm. The particle sizes of cassiterites from XA-3 (350 × 200 μm to 550 × 380 μm) and XA-4 (280 × 240 μm to 490 × 330 μm) were larger than those from XA-5 (130 × 110 μm to 270 × 180 μm).

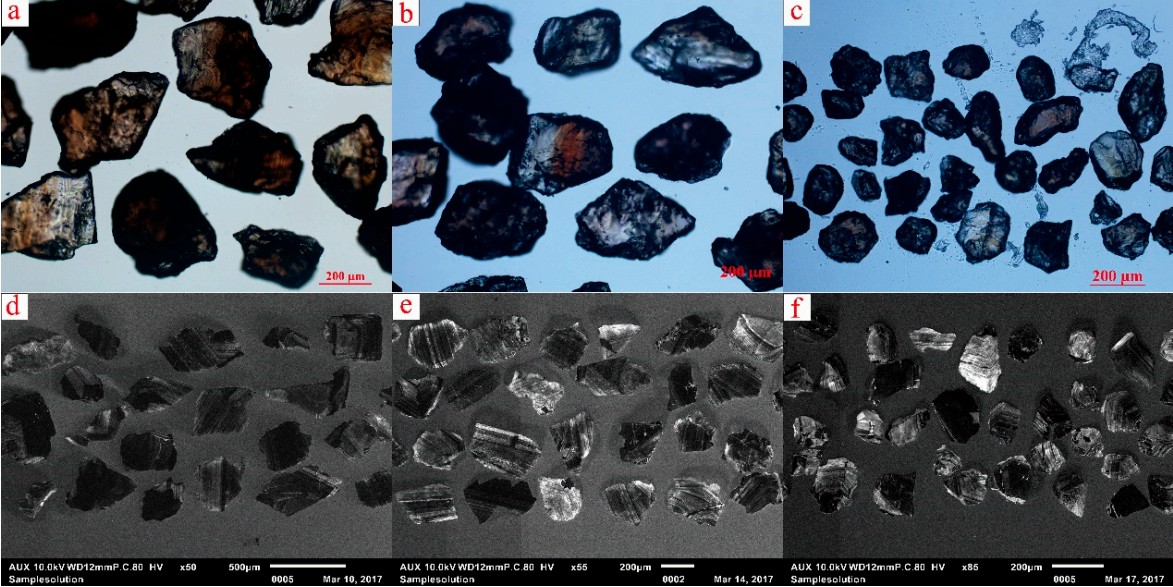

**Figure 4.** Photomicrographs and cathodoluminescence (CL) image of tin ores from the Xi'ao Cu-Sn deposit. XA-3: (**a**,**d**); XA-4: (**b**,**e**); XA-5: (**c**,**f**).

Pronounced differences in the luminescence and internal structures of CL images were observed among cassiterites that formed in different crystallization environments (Figure 5). Based on the luminescence and complexity of the internal structures observed in CL images, the cassiterites could be divided into two zones: (1) the H zone (homogeneous zone), which is characterized by black luminescence and homogeneous internal structures (Figure 5a,b,d), and (2) the O zone (oscillatory zone), which exhibits gray, off-white luminescence with some hourglass structures and oscillatory zoning (Figure 5a,b) and others homogeneous (Figure 5c,e). XA-3 cassiterites had both H and O units, and oscillatory zoning was common in O units. XA-4 cassiterites were predominantly composed of O

units with no H units. XA-5 cassiterites were primarily composed of O units with oscillatory zoning and few H units.

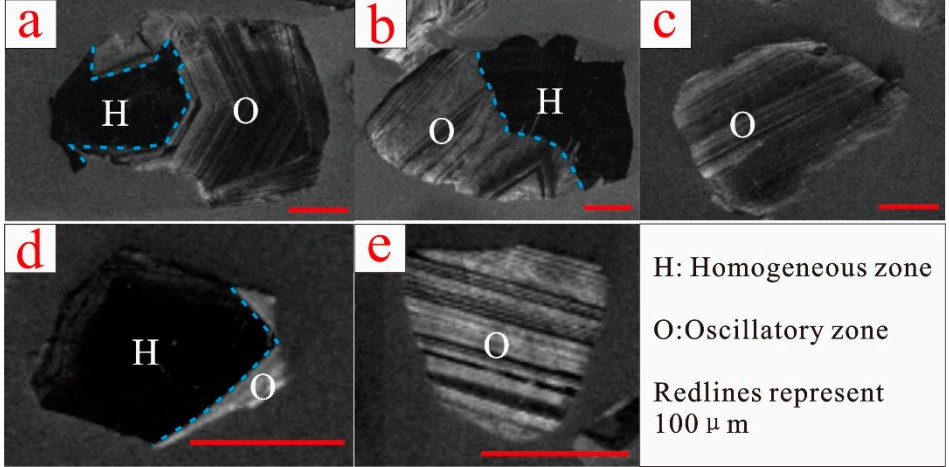

**Figure 5.** Representative CL images of cassiterites from different types of ores in the Xi'ao deposit. XA-3: (**a**,**b**); XA-4: (**c**); XA-5: (**d**,**e**).

### 5.2. Raman Spectrum

Cassiterite has a rutile structure with space group $D_{4h}^{14}$-P4/mnm ($Z = 2$). The tin atom is located in the octahedral coordination of oxygen. The mechanical representation of the normal vibration modes at the center of the Brillouin zone is given as $\Gamma = A_{1g} + A_{2g} + B_{1g} + B_{2g} + E_g + 2B_{1u} + A_{2u} + 3E_u$ [69,70], among which $A_{1g}$, $B_{1g}$, $B_{2g}$ and $E_g$ were assigned to Raman active while $A_{2g} + 2B_{1u}$ was assigned to Raman inactive.

The Raman spectra of Xi'ao cassiterite samples are shown in Figure 6. The most intense peak can be attributed to $A_{1g}$ mode, which was found to shift to 633.693 cm$^{-1}$ in XA-3, 632.295 cm$^{-1}$ in XA-4 and 633.693 cm$^{-1}$ in XA-4, while those exhibited at 473.355 cm$^{-1}$ to 531.539 cm$^{-1}$ and 774.156 cm$^{-1}$ to 776.176 cm$^{-1}$ may have been due to vibrational modes $E_g$ and $B_{2g}$, respectively. $B_{1g}$ mode was found to shift to 88.731 cm$^{-1}$ in XA-3, 87.346 cm$^{-1}$ in XA-4 and 85.058 cm$^{-1}$ in XA-5. The Raman spectra peaks of cassiterites collected from Xi'ao are in good agreement with the pure SnO$_2$ Raman spectra peaks of $A_{1g} = 646$ cm$^{-1}$, $B_{2g} = 752$ cm$^{-1}$, $E_g = 441$ cm$^{-1}$ and $B_{1g} = 100$ cm$^{-1}$ reported by Katiyar et al. [71].

Additional peaks were observed at 438.109 in XA-3 and XA-5 and 233.004 in XA-4, which may have been due to the nano-inclusions, possibly requiring further study.

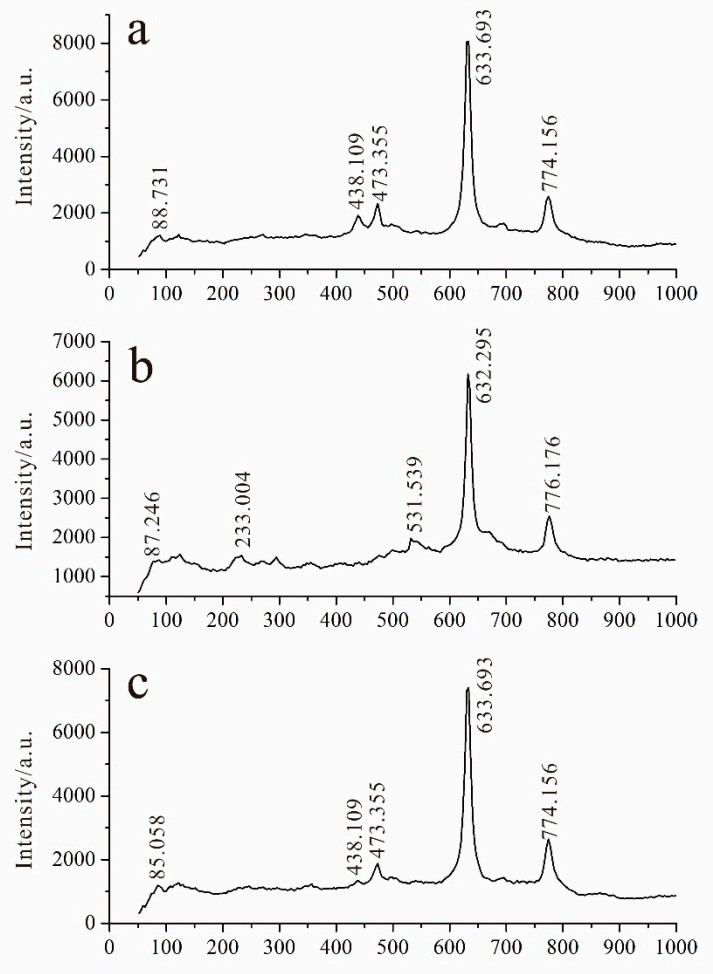

**Figure 6.** Measured Raman spectra of different cassiterite samples. a is XA-3, b is XA-4, c is XA-5.

### 5.3. Cassiterite Composition

Trace element contents of cassiterite are listed in Table 3. The average $SnO_2$ contents of XA-3, XA-4, and XA-5 in the study area were 99.170 wt % (96.631 wt % to 100.416 wt %), 99.037 wt % (96.516 wt % to 100.022 wt %), and 99.338 wt % (98.699 wt % to 99.885 wt %), respectively. The main trace elements in these different types of cassiterite comprised Fe, Nb, Ta, and Ti, and minor trace elements included W, Mn, and Si (Table 3). A summary of the EPMA data is shown in Table 4. Iron was the most abundant minor element in cassiterites (0.498 wt % to 1.181 wt %). In XA-3 and XA-4, Nb abundance was higher than Ta, while in XA-5, Ta abundance was higher. Ti was an important element in cassiterite and contents ranged from 0 to 1.448 wt %.

**Table 3.** Electron probe micro-analyzer (EPMA) data on cassiterites from Xi'ao Cu-Sn polymetallic deposit.

| | XA-3-1 | XA-3-2 | XA-3-4 | XA-5-6 | XA-3-7 | XA-3-8 | XA-3-9 | XA-3-10 | XA-3-11 | XA-3-12 | XA-3-13 | XA-3-15 | XA-3-17 | XA-3-18 | XA-3-20 |
|---|---|---|---|---|---|---|---|---|---|---|---|---|---|---|---|
| | | | | | | | **XA-3 Altered Rock-Type Ore** | | | | | | | | |
| | H | H | O | O | O | O | O | O | O | O | O | O | O | H | H |
| $SnO_2$ | 100.416 | 98.882 | 99.667 | 99.445 | 99.716 | 98.754 | 99.966 | 99.775 | 96.631 | 98.772 | 98.278 | 98.957 | 99.815 | 100.087 | 98.392 |
| $WO_3$ | 0.000 | 0.667 | 0.047 | 0.487 | 0.387 | 0.017 | 0.161 | 0.080 | 0.082 | 0.031 | 0.030 | 0.029 | 0.000 | 0.000 | 0.037 |
| $Nb_2O_5$ | 0.000 | 0.058 | 0.192 | 0.000 | 0.024 | 0.038 | 0.010 | 0.039 | 0.558 | 0.019 | 0.005 | 0.063 | 0.039 | 0.029 | 0.592 |
| $Ta_2O_5$ | 0.000 | 0.014 | 0.255 | 0.000 | 0.000 | 0.277 | 0.000 | 0.213 | 0.000 | 0.000 | 0.000 | 0.047 | 0.000 | 0.000 | 0.110 |
| $FeO$ | 0.625 | 0.782 | 0.626 | 0.744 | 0.678 | 0.501 | 0.498 | 0.530 | 0.712 | 1.168 | 0.952 | 0.721 | 0.876 | 0.653 | 0.621 |
| $MnO$ | 0.023 | 0.032 | 0.015 | 0.024 | 0.000 | 0.016 | 0.000 | 0.006 | 0.002 | 0.005 | 0.000 | 0.000 | 0.000 | 0.002 | 0.000 |
| $TiO_2$ | 0.000 | 0.018 | 0.211 | 0.150 | 0.240 | 0.000 | 0.000 | 0.145 | 0.810 | 0.209 | 1.052 | 0.772 | 0.235 | 0.219 | 0.638 |
| $SiO_2$ | 0.012 | 0.005 | 0.080 | 0.024 | 0.011 | 0.041 | 0.004 | 0.047 | 0.054 | 0.000 | 0.000 | 0.026 | 0.000 | 0.031 | 0.000 |
| Total | 101.076 | 100.458 | 101.093 | 100.874 | 101.056 | 99.644 | 100.639 | 100.835 | 98.849 | 100.204 | 100.317 | 100.615 | 100.965 | 101.021 | 100.390 |
| | | | | | | | **Cation Formula Based on Four Atoms of Oxygen** | | | | | | | | |
| $Sn^{4+}$ | 1.9795 | 1.9593 | 1.9580 | 1.9603 | 1.9615 | 1.9798 | 1.9849 | 1.9737 | 1.9327 | 1.9639 | 1.9383 | 1.9504 | 1.9700 | 1.9745 | 1.9428 |
| $W^{6+}$ | 0.0000 | 0.0086 | 0.0006 | 0.0062 | 0.0049 | 0.0002 | 0.0021 | 0.0010 | 0.0011 | 0.0004 | 0.0004 | 0.0004 | 0.0000 | 0.0000 | 0.0005 |
| $Nb^{5+}$ | 0.0000 | 0.0013 | 0.0043 | 0.0000 | 0.0005 | 0.0009 | 0.0002 | 0.0009 | 0.0126 | 0.0004 | 0.0001 | 0.0014 | 0.0009 | 0.0006 | 0.0133 |
| $Ta^{5+}$ | 0.0000 | 0.0002 | 0.0034 | 0.0000 | 0.0000 | 0.0038 | 0.0000 | 0.0029 | 0.0000 | 0.0000 | 0.0000 | 0.0006 | 0.0000 | 0.0000 | 0.0015 |
| $Fe^{2+}$ | 0.0388 | 0.0488 | 0.0387 | 0.0462 | 0.0420 | 0.0351 | 0.0346 | 0.0367 | 0.0498 | 0.0812 | 0.0657 | 0.0497 | 0.0605 | 0.0451 | 0.0429 |
| $Mn^{2+}$ | 0.0010 | 0.0013 | 0.0006 | 0.0010 | 0.0000 | 0.0007 | 0.0000 | 0.0003 | 0.0001 | 0.0002 | 0.0000 | 0.0000 | 0.0000 | 0.0001 | 0.0000 |
| $Ti^{4+}$ | 0.0000 | 0.0007 | 0.0078 | 0.0056 | 0.0089 | 0.0000 | 0.0000 | 0.0054 | 0.0306 | 0.0078 | 0.0391 | 0.0287 | 0.0087 | 0.0081 | 0.0238 |
| $Si^{4+}$ | 0.0006 | 0.0003 | 0.0039 | 0.0012 | 0.0005 | 0.0020 | 0.0002 | 0.0023 | 0.0027 | 0.0000 | 0.0000 | 0.0013 | 0.0000 | 0.0015 | 0.0000 |
| Total | 2.0199 | 2.0204 | 2.0175 | 2.0205 | 2.0184 | 2.0225 | 2.0219 | 2.0232 | 2.0295 | 2.0540 | 2.0436 | 2.0324 | 2.0401 | 2.0299 | 2.0247 |
| Nb + Ta | 0.0000 | 0.0015 | 0.0077 | 0.0000 | 0.0005 | 0.0047 | 0.0002 | 0.0037 | 0.0126 | 0.0004 | 0.0001 | 0.0020 | 0.0009 | 0.0006 | 0.0147 |
| Fe + Mn | 0.0398 | 0.0501 | 0.0393 | 0.0472 | 0.0420 | 0.0358 | 0.0346 | 0.0370 | 0.0499 | 0.0814 | 0.0657 | 0.0497 | 0.0605 | 0.0452 | 0.0429 |

| | XA-4-1 | XA-4-2 | XA-4-3 | XA-4-4 | XA-4-6 | XA-4-7 | XA-4-8 | XA-4-9 | XA-4-5 | XA-4-10 | XA-4-11 | XA-4-12 | XA-4-13 | XA-4-14 | XA-4-15 | XA-4-16 | XA-4-17 |
|---|---|---|---|---|---|---|---|---|---|---|---|---|---|---|---|---|---|
| | | | | | | | | **XA-4 Altered Rock-Type Ore** | | | | | | | | | |
| | O | O | O | H | H | H | H | H | | | | | | | H | | |
| $SnO_2$ | 98.483 | 99.626 | 99.369 | 99.397 | 99.949 | 99.423 | 98.811 | 99.598 | 96.516 | 98.760 | 99.027 | 98.777 | 99.634 | 99.005 | 98.994 | 98.236 | 100.022 |
| $WO_3$ | 0.062 | 0.065 | 0.000 | 0.052 | 0.011 | 0.000 | 0.037 | 0.000 | 0.087 | 0.143 | 0.000 | 0.000 | 0.042 | 0.000 | 0.000 | 0.547 | 0.012 |
| $Nb_2O_5$ | 0.029 | 0.000 | 0.039 | 0.000 | 0.106 | 0.067 | 0.000 | 0.082 | 0.513 | 0.250 | 0.005 | 0.000 | 0.000 | 0.086 | 0.000 | 0.077 | 0.038 |
| $Ta_2O_5$ | 0.047 | 0.077 | 0.000 | 0.000 | 0.000 | 0.019 | 0.000 | 0.049 | 0.339 | 0.184 | 0.000 | 0.025 | 0.000 | 0.000 | 0.118 | 0.093 | 0.000 |
| $FeO$ | 0.614 | 0.680 | 0.965 | 0.597 | 0.521 | 0.800 | 0.795 | 0.553 | 0.959 | 0.588 | 0.587 | 0.751 | 0.656 | 0.606 | 0.802 | 0.606 | 0.590 |
| $MnO$ | 0.028 | 0.000 | 0.028 | 0.000 | 0.007 | 0.000 | 0.000 | 0.000 | 0.000 | 0.019 | 0.011 | 0.000 | 0.015 | 0.000 | 0.014 | 0.000 | 0.000 |
| $TiO_2$ | 0.400 | 0.381 | 0.308 | 0.066 | 0.282 | 0.014 | 0.036 | 0.000 | 1.448 | 0.225 | 0.754 | 0.392 | 0.179 | 0.628 | 0.632 | 0.849 | 0.201 |
| $SiO_2$ | 0.048 | 0.019 | 0.032 | 0.033 | 0.030 | 0.057 | 0.064 | 0.003 | 0.064 | 0.028 | 0.021 | 0.024 | 0.043 | 0.021 | 0.000 | 0.016 | 0.048 |
| Total | 99.711 | 100.848 | 100.741 | 100.145 | 100.906 | 100.380 | 99.743 | 100.285 | 99.926 | 100.197 | 100.405 | 99.969 | 100.569 | 100.346 | 100.560 | 100.424 | 100.911 |
| | | | | | | | | **Cation Formula Based on Four Atoms of Oxygen** | | | | | | | | | |
| $Sn^{4+}$ | 1.9648 | 1.9667 | 1.9631 | 1.9810 | 1.9727 | 1.9759 | 1.9762 | 1.9838 | 1.8733 | 1.9632 | 1.9570 | 1.9662 | 1.9749 | 1.9591 | 1.9557 | 1.9394 | 1.9753 |
| $W^{6+}$ | 0.0008 | 0.0008 | 0.0000 | 0.0007 | 0.0001 | 0.0000 | 0.0005 | 0.0000 | 0.0011 | 0.0018 | 0.0000 | 0.0000 | 0.0005 | 0.0000 | 0.0000 | 0.0070 | 0.0002 |
| $Nb^{5+}$ | 0.0007 | 0.0000 | 0.0009 | 0.0000 | 0.0024 | 0.0015 | 0.0000 | 0.0018 | 0.0121 | 0.0056 | 0.0001 | 0.0000 | 0.0000 | 0.0019 | 0.0000 | 0.0017 | 0.0009 |
| $Ta^{5+}$ | 0.0006 | 0.0010 | 0.0000 | 0.0000 | 0.0000 | 0.0003 | 0.0000 | 0.0007 | 0.0045 | 0.0025 | 0.0000 | 0.0003 | 0.0000 | 0.0000 | 0.0016 | 0.0013 | 0.0000 |
| $Fe^{2+}$ | 0.0428 | 0.0470 | 0.0667 | 0.0416 | 0.0360 | 0.0556 | 0.0556 | 0.0386 | 0.0651 | 0.0409 | 0.0405 | 0.0523 | 0.0455 | 0.0419 | 0.0554 | 0.0418 | 0.0408 |
| $Mn^{2+}$ | 0.0012 | 0.0000 | 0.0012 | 0.0000 | 0.0005 | 0.0000 | 0.0000 | 0.0000 | 0.0000 | 0.0008 | 0.0005 | 0.0000 | 0.0006 | 0.0000 | 0.0006 | 0.0000 | 0.0000 |
| $Ti^{4+}$ | 0.0151 | 0.0142 | 0.0115 | 0.0025 | 0.0105 | 0.0005 | 0.0013 | 0.0000 | 0.0530 | 0.0084 | 0.0281 | 0.0147 | 0.0067 | 0.0234 | 0.0235 | 0.0316 | 0.0075 |
| $Si^{4+}$ | 0.0024 | 0.0009 | 0.0016 | 0.0017 | 0.0015 | 0.0028 | 0.0032 | 0.0002 | 0.0031 | 0.0014 | 0.0010 | 0.0012 | 0.0021 | 0.0010 | 0.0000 | 0.0008 | 0.0024 |

| | | | | | | | | | | | | | | | | | |
|---|---|---|---|---|---|---|---|---|---|---|---|---|---|---|---|---|---|
| Total | 2.0284 | 2.0306 | 2.0448 | 2.0274 | 2.0234 | 2.0366 | 2.0368 | 2.0251 | 2.0122 | 2.0247 | 2.0272 | 2.0348 | 2.0303 | 2.0274 | 2.0368 | 2.0236 | 2.0269 |
| Nb + Ta | 0.0013 | 0.0010 | 0.0009 | 0.0000 | 0.0024 | 0.0018 | 0.0000 | 0.0025 | 0.0166 | 0.0081 | 0.0001 | 0.0003 | 0.0000 | 0.0019 | 0.0016 | 0.0030 | 0.0009 |
| Fe + Mn | 0.0440 | 0.0470 | 0.0678 | 0.0416 | 0.0362 | 0.0556 | 0.0556 | 0.0386 | 0.0651 | 0.0417 | 0.0410 | 0.0523 | 0.0461 | 0.0419 | 0.0560 | 0.0418 | 0.0408 |

| XA-5 Tourmaline Vein-Type Ore | | | | | | | | | | | | | | | | |
|---|---|---|---|---|---|---|---|---|---|---|---|---|---|---|---|---|
| | XA-5-6 | XA-5-8 | XA-5-9 | XA-5-10 | XA-5-11 | XA-5-12 | XA-5-13 | XA-5-15 | XA-5-16 | XA-5-17 | XA-5-18 | XA-5-19 | XA-5-20 | XA-5-21 | XA-5-22 | XA-5-23 |
| | O | O | O | O | H | O | O | O | O | O | O | O | O | O | O | O |
| $SnO_2$ | 99.885 | 99.022 | 99.149 | 99.757 | 99.161 | 99.584 | 99.124 | 98.823 | 99.568 | 98.975 | 99.168 | 99.867 | 99.452 | 98.699 | 99.362 | 99.816 |
| $WO_3$ | 0.000 | 0.031 | 0.128 | 0.000 | 0.033 | 0.000 | 0.016 | 0.066 | 0.000 | 0.070 | 0.062 | 0.000 | 0.046 | 0.000 | 0.035 | 0.000 |
| $Nb_2O_5$ | 0.082 | 0.000 | 0.067 | 0.048 | 0.048 | 0.043 | 0.149 | 0.000 | 0.000 | 0.000 | 0.034 | 0.000 | 0.111 | 0.000 | 0.115 | 0.024 |
| $Ta_2O_5$ | 0.000 | 0.000 | 0.066 | 0.000 | 0.376 | 0.003 | 0.334 | 0.123 | 0.060 | 0.189 | 0.104 | 0.000 | 0.000 | 0.044 | 0.305 | 0.071 |
| FeO | 0.655 | 0.666 | 0.723 | 0.543 | 1.081 | 0.645 | 0.643 | 1.040 | 0.705 | 0.991 | 1.082 | 1.181 | 0.588 | 1.065 | 0.857 | 0.649 |
| MnO | 0.034 | 0.000 | 0.011 | 0.000 | 0.000 | 0.009 | 0.000 | 0.000 | 0.009 | 0.012 | 0.000 | 0.006 | 0.035 | 0.017 | 0.000 | 0.010 |
| $TiO_2$ | 0.000 | 0.216 | 0.070 | 0.098 | 0.135 | 0.154 | 0.237 | 0.188 | 0.177 | 0.005 | 0.021 | 0.000 | 0.095 | 0.085 | 0.202 | 0.154 |
| $SiO_2$ | 0.028 | 0.000 | 0.040 | 0.000 | 0.012 | 0.041 | 0.037 | 0.030 | 0.077 | 0.068 | 0.033 | 0.070 | 0.053 | 0.025 | 0.025 | 0.011 |
| Total | 100.684 | 99.935 | 100.254 | 100.446 | 100.846 | 100.479 | 100.540 | 100.270 | 100.596 | 100.310 | 100.504 | 101.124 | 100.380 | 99.935 | 100.901 | 100.735 |
| Cation Formula Based on Four Atoms of Oxygen | | | | | | | | | | | | | | | | |
| $Sn^{4+}$ | 1.9752 | 1.9704 | 1.9671 | 1.9780 | 1.9526 | 1.9707 | 1.9592 | 1.9558 | 1.9665 | 1.9608 | 1.9600 | 1.9584 | 1.9707 | 1.9612 | 1.9557 | 1.9715 |
| $W^{6+}$ | 0.0000 | 0.0004 | 0.0016 | 0.0000 | 0.0004 | 0.0000 | 0.0002 | 0.0008 | 0.0000 | 0.0009 | 0.0008 | 0.0000 | 0.0006 | 0.0000 | 0.0004 | 0.0000 |
| $Nb^{5+}$ | 0.0018 | 0.0000 | 0.0015 | 0.0011 | 0.0011 | 0.0010 | 0.0033 | 0.0000 | 0.0000 | 0.0000 | 0.0008 | 0.0000 | 0.0025 | 0.0000 | 0.0026 | 0.0005 |
| $Ta^{5+}$ | 0.0000 | 0.0000 | 0.0009 | 0.0000 | 0.0050 | 0.0000 | 0.0045 | 0.0017 | 0.0008 | 0.0026 | 0.0014 | 0.0000 | 0.0000 | 0.0006 | 0.0041 | 0.0010 |
| $Fe^{2+}$ | 0.0408 | 0.0417 | 0.0452 | 0.0339 | 0.0670 | 0.0402 | 0.0401 | 0.0648 | 0.0439 | 0.0618 | 0.0673 | 0.0751 | 0.0367 | 0.0666 | 0.0531 | 0.0403 |
| $Mn^{2+}$ | 0.0014 | 0.0000 | 0.0005 | 0.0000 | 0.0000 | 0.0004 | 0.0000 | 0.0000 | 0.0004 | 0.0005 | 0.0000 | 0.0003 | 0.0015 | 0.0007 | 0.0000 | 0.0004 |
| $Ti^{4+}$ | 0.0000 | 0.0081 | 0.0026 | 0.0037 | 0.0050 | 0.0057 | 0.0088 | 0.0070 | 0.0066 | 0.0002 | 0.0008 | 0.0000 | 0.0036 | 0.0032 | 0.0075 | 0.0057 |
| $Si^{4+}$ | 0.0014 | 0.0000 | 0.0020 | 0.0000 | 0.0006 | 0.0021 | 0.0018 | 0.0015 | 0.0038 | 0.0034 | 0.0017 | 0.0037 | 0.0026 | 0.0013 | 0.0013 | 0.0006 |
| Total | 2.0207 | 2.0206 | 2.0214 | 2.0167 | 2.0318 | 2.0200 | 2.0180 | 2.0315 | 2.0219 | 2.0301 | 2.0327 | 2.0374 | 2.0182 | 2.0335 | 2.0246 | 2.0200 |
| Nb + Ta | 0.0018 | 0.0000 | 0.0024 | 0.0011 | 0.0061 | 0.0010 | 0.0078 | 0.0017 | 0.0008 | 0.0026 | 0.0022 | 0.0000 | 0.0025 | 0.0006 | 0.0067 | 0.0015 |
| Fe + Mn | 0.0422 | 0.0417 | 0.0456 | 0.0339 | 0.0670 | 0.0406 | 0.0401 | 0.0648 | 0.0442 | 0.0623 | 0.0673 | 0.0754 | 0.0382 | 0.0673 | 0.0531 | 0.0407 |

**Table 4.** Statistical characteristics of EPMA results.

| Sample | SnO$_2$ | WO$_3$ | Nb$_2$O$_5$ | Ta$_2$O$_5$ | FeO | MnO | TiO$_2$ | SiO$_2$ |
|---|---|---|---|---|---|---|---|---|
| XA-3 | 99.170 ± 0.95[a] | 0.137 ± 0.21 | 0.111 ± 0.19 | 0.061 ± 0.1 | 0.713 ± 0.18 | 0.008 ± 0.01 | 0.313 ± 0.34 | 0.022 ± 0.02 |
|  | 96.631~100.416[b] | 0–0.667 | 0–0.592 | 0–0.277 | 0.498–1.168 | 0–0.032 | 0–1.052 | 0–0.080 |
| XA-4 | 99.037 ± 0.82 | 0.062 ± 0.13 | 0.076 ± 0.13 | 0.056 ± 0.09 | 0.686 ± 0.14 | 0.007 ± 0.01 | 0.400 ± 0.37 | 0.032 ± 0.02 |
|  | 96.516–100.022 | 0–0.547 | 0–0.513 | 0–0.339 | 0.521–0.964 | 0–0.028 | 0–1.448 | 0–0.064 |
| XA-5 | 99.338 ± 0.38 | 0.030 ± 0.04 | 0.045 ± 0.05 | 0.105 ± 0.13 | 0.820 ± 0.22 | 0.009 ± 0.01 | 0.115 ± 0.08 | 0.034 ± 0.02 |
|  | 98.699–99.885 | 0–0.128 | 0–0.149 | 0–0.376 | 0.543–1.181 | 0–0.035 | 0–0.237 | 0–0.077 |

[a]: mean ± sd, [b]: range of concentration.

*5.4. Oxygen Isotopes*

The results of oxygen isotope determination of the cassiterite samples are shown in Table 5. The cassiterite samples showed δ$^{18}$O values ranging from 3.9‰ to 4.7‰ (Table 5), exhibiting a gradual decrease from XA-3 to XA-4 to XA-5. The δ$^{18}$O values of XA-3, XA-4 and XA-5 were 4.7‰, 4.4‰ and 3.9‰, respectively. Calculating the precise δ$^{18}$O$_{H2O}$ values of fluids that were in equilibrium with cassiterite was difficult due to the large variation in temperatures obtained from fluid inclusions [50]. Here, we used the average homogenization temperatures of these fluid inclusions to calculate their δ$^{18}$O$_{H2O}$ values. The calculated δ$^{18}$O$_{H2O}$ values of these fluids were similar and within a limited range (7.16‰ to 8.25‰). The δ$^{18}$O$_{H2O}$ values of XA-3, XA-4, and XA-5 were 8.25‰, 7.95‰, and 7.16‰, respectively.

**Table 5.** The δ$^{18}$O values of cassiterite samples in Xi'ao Cu-Sn polymetallic deposit.

| Samples | Minerals | δ$^{18}$O$_{V-PDB}$ (‰) | δ$^{18}$O$_{V-SMOW}$ (‰) | Temperature (°C) [a] | δ$^{18}$O$_{H2O-SMOW}$ (‰) |
|---|---|---|---|---|---|
| XA-3 |  | −25.40 | 4.7 | 374.10 | 8.25 |
| XA-4 | Cassiterite | −25.70 | 4.4 | 374.10 | 7.95 |
| XA-5 |  | −26.20 | 3.9 | 353.82 | 7.16 |

[a], temperatures were taken from [50].

*5.5. U-Pb Ages*

The cassiterite grains used for U-Pb isotope analysis had a few cracks and fluid inclusions. Care was taken when selecting laser positions to avoid cracks and fluid inclusions to reduce the influence of common Pb in inclusions and to improve the accuracy of analyses. Both O and H zones were used for U-Pb dating. The LA-ICP-MS results are summarized in Table 6 and plotted in Figure 7. Figure 7 indicates that cassiterites from the XA-3, XA-4, and XA-5 yielded U-Pb Tera-Wasserburg concordia lower intercept ages of 83.3 ± 2.1 Ma (1σ, MSWD = 0.29) (MSWD: mean squares weighted deviates), 84.9 ± 1.7 Ma (1σ, MSWD = 0.78), and 84.0 ± 5.6 Ma (1σ, MSWD = 4.1), respectively. These data represent the metallogenic age of the Xi'ao Cu-Sn polymetallic deposit.

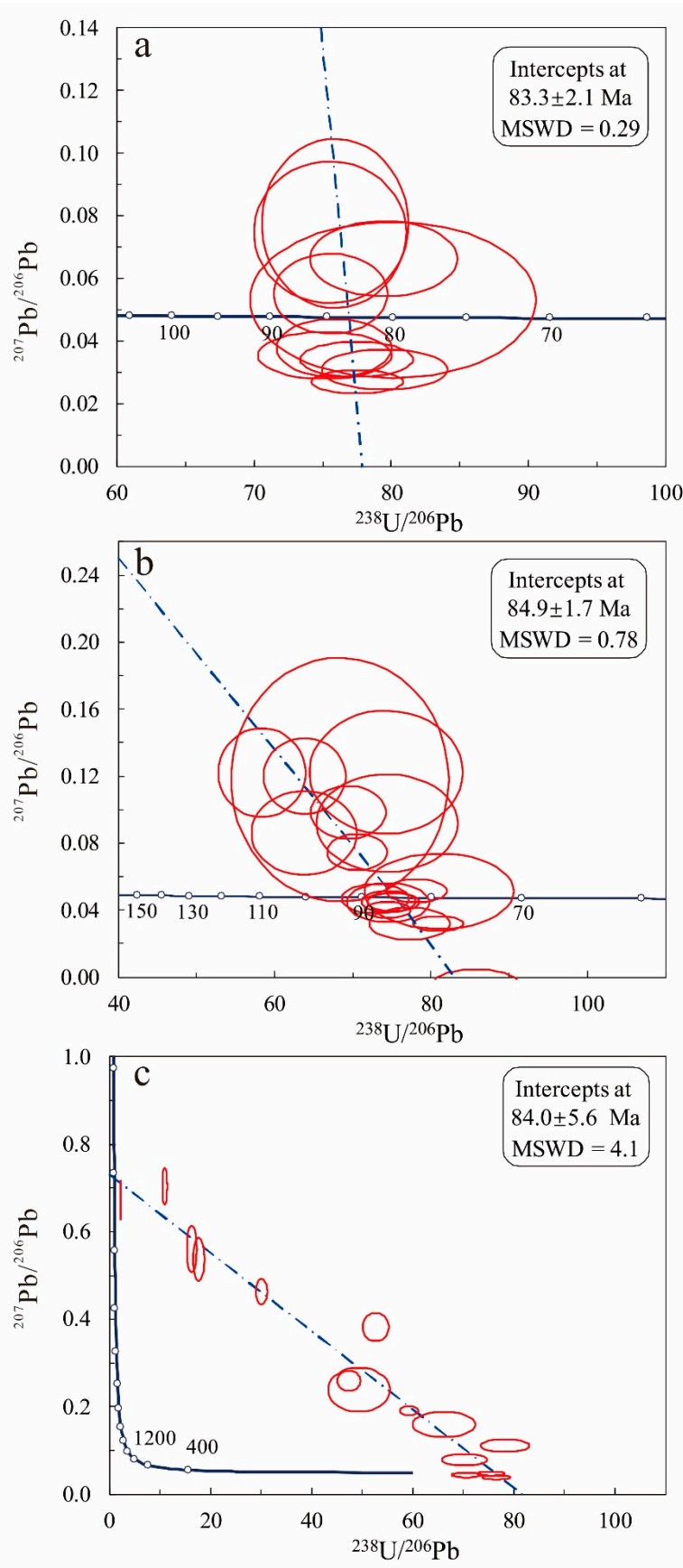

**Figure 7.** U-Pb Tera-Wasserburg concordia age for cassiterite samples from Xi'ao deposit. a is the age of XA-3, b is the age XA-4, c is the age XA-5.

**Table 6.** LA-ICP-MS U-Pb dating results of cassiterite from the Xi'ao Cu-Sn polymetallic deposit.

**XA-3 Altered Rock-Type Ore**

| Spots | Th | U | Isotopic Ratios | | | | | | | | Age (Ma) | | | | | |
| | | | $^{238}U/^{206}Pb$ | 1σ | $^{207}Pb/^{206}Pb$ | 1σ | $^{206}Pb/^{238}U$ | 1σ | $^{207}Pb/^{235}U$ | 1σ | $^{207}Pb/^{206}Pb$ | 1σ | $^{206}Pb/^{238}U$ | 1σ | $^{207}Pb/^{235}U$ | 1σ |
|---|---|---|---|---|---|---|---|---|---|---|---|---|---|---|---|---|
| XA16-3-3 | 0 | 20,627 | 75.1314801 | 3.217501 | 0.03543 | 0.00497 | 0.01331 | 0.00057 | 0.06532 | 0.00877 | 0.1 | 0 | 85.3 | 3.62 | 64.2 | 8.36 |
| XA16-3-4 | 8 | 64,216 | 77.5193798 | 2.223424 | 0.02693 | 0.00255 | 0.0129 | 0.00037 | 0.04806 | 0.00438 | 0.1 | 0 | 82.6 | 2.33 | 47.7 | 4.24 |
| XA16-3-5 | 0 | 38,135 | 77.4593338 | 2.699977 | 0.03421 | 0.00376 | 0.01291 | 0.00045 | 0.06101 | 0.00639 | 0.1 | 0 | 82.7 | 2.86 | 60.1 | 6.11 |
| XA16-3-6 | 0 | 32,544 | 79.491256 | 3.033053 | 0.031 | 0.00423 | 0.01258 | 0.00048 | 0.05385 | 0.00708 | 0.1 | 0 | 80.6 | 3.07 | 53.3 | 6.82 |
| XA16-3-8 | 0 | 23,724 | 79.4281176 | 3.532942 | 0.06637 | 0.00797 | 0.01259 | 0.00056 | 0.11525 | 0.01289 | 818.2 | 232.61 | 80.7 | 3.55 | 110.8 | 11.74 |
| XA16-3-11 | 0 | 46,501 | 75.8150114 | 2.759 | 0.0378 | 0.00616 | 0.01319 | 0.00048 | 0.06922 | 0.01094 | 0.1 | 0 | 84.5 | 3.08 | 68 | 10.39 |
| XA16-3-12 | 0 | 51,406 | 80.1282051 | 6.869966 | 0.05307 | 0.01635 | 0.01248 | 0.00107 | 0.09228 | 0.02726 | 331.9 | 579.52 | 80 | 6.84 | 89.6 | 25.34 |
| XA16-3-13 | 0 | 64,041 | 75.5857899 | 2.742342 | 0.05508 | 0.00849 | 0.01323 | 0.00048 | 0.102 | 0.01516 | 415.3 | 312.01 | 84.7 | 3.08 | 98.6 | 13.97 |
| XA16-3-14 | 0 | 23,742 | 75.5287009 | 3.650934 | 0.07457 | 0.01498 | 0.01324 | 0.00064 | 0.13904 | 0.02702 | 1056.5 | 359.19 | 84.8 | 4.08 | 132.2 | 24.09 |
| XA16-3-16 | 0 | 32,855 | 75.8725341 | 3.511551 | 0.07766 | 0.01786 | 0.01318 | 0.00061 | 0.14698 | 0.0334 | 1138.3 | 400.23 | 84.4 | 3.86 | 139.2 | 29.57 |

**XA-4 Altered Rock-Type Ore**

| Spots | Th | U | Isotopic Ratios | | | | | | | | Ages (Ma) | | | | | |
| | | | $^{238}U/^{206}Pb$ | 1σ | $^{207}Pb/^{206}Pb$ | 1σ | $^{206}Pb/^{238}U$ | 1σ | $^{207}Pb/^{235}U$ | 1σ | $^{207}Pb/^{206}Pb$ | 1σ | $^{206}Pb/^{238}U$ | 1σ | $^{207}Pb/^{235}U$ | 1σ |
|---|---|---|---|---|---|---|---|---|---|---|---|---|---|---|---|---|
| XA-4-1 | 0 | 38,164 | 70.52186 | 2.5364 | 0.07462 | 0.00723 | 0.01418 | 0.00051 | 0.14623 | 0.0133 | 1058 | 183.86 | 90.8 | 3.26 | 138.6 | 11.78 |
| XA-4-2 | 1 | 36,083 | 69.39625 | 3.178454 | 0.09839 | 0.01033 | 0.01441 | 0.00066 | 0.19591 | 0.01873 | 1593.9 | 184.1 | 92.2 | 4.18 | 181.7 | 15.9 |
| XA-4-3 | 3 | 91,084 | 74.46016 | 1.940511 | 0.04016 | 0.00318 | 0.01343 | 0.00035 | 0.07453 | 0.00565 | 0.1 | 0 | 86 | 2.26 | 73 | 5.34 |
| XA-4-4 | 0 | 31,407 | 72.83321 | 2.811479 | 0.0469 | 0.00583 | 0.01373 | 0.00053 | 0.08894 | 0.01058 | 44 | 273.39 | 87.9 | 3.4 | 86.5 | 9.86 |
| XA-4-5 | 0 | 58,372 | 85.6898 | 3.597944 | −0.00366 | 0.00568 | 0.01167 | 0.00049 | −0.0059 | 0.00916 | 0.1 | 0 | 74.8 | 3.13 | −6 | 9.35 |
| XA-4-6 | 0 | 72,882 | 74.51565 | 2.276559 | 0.04511 | 0.00408 | 0.01342 | 0.00041 | 0.08358 | 0.00721 | 0.1 | 157.16 | 85.9 | 2.62 | 81.5 | 6.75 |
| XA-4-7 | 0 | 18,842 | 74.46016 | 3.492919 | 0.04477 | 0.00769 | 0.01343 | 0.00063 | 0.08302 | 0.01377 | 0.1 | 305.34 | 86 | 3.99 | 81 | 12.91 |
| XA-4-8 | 0 | 6665 | 63.77551 | 4.392701 | 0.08595 | 0.01659 | 0.01568 | 0.00108 | 0.18605 | 0.03366 | 1336.9 | 333.6 | 100.3 | 6.88 | 173.3 | 28.81 |
| XA-4-9 | 0 | 11,272 | 63.85696 | 3.466055 | 0.12024 | 0.01501 | 0.01566 | 0.00085 | 0.25995 | 0.02943 | 1959.8 | 207.41 | 100.2 | 5.42 | 234.6 | 23.72 |
| XA-4-10 | 0 | 7160 | 58.34306 | 3.676225 | 0.12216 | 0.01768 | 0.01714 | 0.00108 | 0.28893 | 0.03792 | 1988 | 237.23 | 109.6 | 6.82 | 257.7 | 29.88 |
| XA-4-11 | 0 | 126,487 | 80.84074 | 2.156625 | 0.03191 | 0.00289 | 0.01237 | 0.00033 | 0.05443 | 0.00475 | 0.1 | 0 | 79.2 | 2.1 | 53.8 | 4.58 |
| XA-4-12 | 0 | 5181 | 74.23905 | 6.448381 | 0.12237 | 0.02421 | 0.01347 | 0.00117 | 0.22732 | 0.04059 | 1991.1 | 315.34 | 86.2 | 7.43 | 208 | 33.58 |
| XA-4-13 | 0 | 58,639 | 78.125 | 2.563477 | 0.05157 | 0.00471 | 0.0128 | 0.00042 | 0.09105 | 0.00785 | 266.4 | 196.22 | 82 | 2.65 | 88.5 | 7.31 |
| XA-4-14 | 0 | 1704 | 68.3527 | 9.1573 | 0.11805 | 0.04805 | 0.01463 | 0.00196 | 0.23816 | 0.09165 | 1926.9 | 591.79 | 93.6 | 12.46 | 216.9 | 75.16 |
| XA-4-15 | 0 | 9525 | 80.97166 | 6.294153 | 0.05115 | 0.01484 | 0.01235 | 0.00096 | 0.08714 | 0.02439 | 247.4 | 558.56 | 79.2 | 6.11 | 84.8 | 22.78 |
| XA-4-16 | 0 | 6612 | 74.34944 | 5.970067 | 0.09207 | 0.01935 | 0.01345 | 0.00108 | 0.17076 | 0.03324 | 1468.7 | 354.03 | 86.1 | 6.9 | 160.1 | 28.83 |
| XA-4-17 | 0 | 97,931 | 75.18797 | 2.20476 | 0.04487 | 0.00372 | 0.0133 | 0.00039 | 0.08227 | 0.00645 | 0.1 | 128.04 | 85.2 | 2.48 | 80.3 | 6.06 |
| XA-4-18 | 0 | 26,046 | 77.27975 | 3.404131 | 0.03189 | 0.0063 | 0.01294 | 0.00057 | 0.0569 | 0.01097 | 0.1 | 0 | 82.9 | 3.64 | 56.2 | 10.54 |

**XA-5 Tourmaline Vein-Type Ore**

| Spots | Th | U | Isotopic Ratios | | | | | | | | Age (Ma) | | | | | |
| | | | $^{238}U/^{206}Pb$ | 1σ | $^{207}Pb/^{206}Pb$ | 1σ | $^{206}Pb/^{238}U$ | 1σ | $^{207}Pb/^{235}U$ | 1σ | $^{207}Pb/^{206}Pb$ | 1σ | $^{206}Pb/^{238}U$ | 1σ | $^{207}Pb/^{235}U$ | 1σ |
|---|---|---|---|---|---|---|---|---|---|---|---|---|---|---|---|---|
| XA-5-1 | 57 | 33,209 | 10.9769484 | 0.277135 | 0.70382 | 0.02819 | 0.0911 | 0.0023 | 8.88783 | 0.30736 | 4738.5 | 56.34 | 562 | 13.59 | 2326.6 | 31.56 |
| XA-5-2 | 1 | 20,863 | 70.2247191 | 2.958907 | 0.07856 | 0.00878 | 0.01424 | 0.0006 | 0.155 | 0.01621 | 1161 | 206.92 | 91.1 | 3.8 | 146.3 | 14.25 |
| XA-5-4 | 35 | 30,390 | 30.0390508 | 0.776016 | 0.46305 | 0.01974 | 0.03329 | 0.00086 | 2.13586 | 0.0787 | 4127.7 | 61.84 | 211.1 | 5.33 | 1160.5 | 25.48 |
| XA-5-5 | 8 | 25,787 | 49.4071146 | 4.027754 | 0.23846 | 0.03313 | 0.02024 | 0.00165 | 0.66883 | 0.07615 | 3109.7 | 205.5 | 129.2 | 10.44 | 520 | 46.33 |
| XA-5-6 | 133 | 143,761 | 59.2417062 | 1.228353 | 0.19086 | 0.00774 | 0.01688 | 0.00035 | 0.44634 | 0.01674 | 2749.6 | 65.1 | 107.9 | 2.2 | 374.7 | 11.75 |
| XA-5-7 | 60 | 27,856 | 17.6584849 | 0.763964 | 0.53733 | 0.03368 | 0.05663 | 0.00245 | 4.21513 | 0.20401 | 4347 | 88.87 | 355.1 | 14.93 | 1677 | 39.72 |
| XA-5-8 | 0 | 125,483 | 75.4716981 | 1.822713 | 0.04799 | 0.00317 | 0.01325 | 0.00032 | 0.0881 | 0.00555 | 97.6 | 150.41 | 84.9 | 2.04 | 85.7 | 5.18 |
| XA-5-9 | 23 | 8254 | 66.0938533 | 4.106294 | 0.15994 | 0.01956 | 0.01513 | 0.00094 | 0.33533 | 0.03576 | 2455.1 | 193.27 | 96.8 | 5.94 | 293.6 | 27.19 |
| XA-5-10 | 74 | 112,650 | 78.1860829 | 3.239924 | 0.11092 | 0.00998 | 0.01279 | 0.00053 | 0.19659 | 0.01594 | 1814.6 | 155.02 | 81.9 | 3.38 | 182.2 | 13.52 |

| XA-5-11 | 44 | 47,919 | 47.3484848 | 1.546897 | 0.25907 | 0.01524 | 0.02112 | 0.00069 | 0.75849 | 0.03861 | 3241 | 89.79 | 134.8 | 4.35 | 573.1 | 22.29 |
|---------|----|--------|------------|----------|---------|---------|---------|---------|---------|---------|--------|--------|--------|-------|--------|-------|
| XA-5-12 | 55 | 32,069 | 52.6315789 | 1.717452 | 0.3831 | 0.02155 | 0.019 | 0.00062 | 1.00882 | 0.04815 | 3844.1 | 82.39 | 121.3 | 3.95 | 708.3 | 24.34 |
| XA-5-13 | 40 | 6225 | 2.12417954 | 0.049588 | 0.67373 | 0.03003 | 0.47077 | 0.01099 | 43.97262 | 1.81795 | 4675.7 | 62.73 | 2486.9 | 48.17 | 3864.6 | 41.05 |
| XA-5-14 | 31 | 5892 | 16.2839928 | 0.654966 | 0.5602 | 0.035 | 0.06141 | 0.00247 | 4.77057 | 0.24063 | 4408 | 88.3 | 384.2 | 15.03 | 1779.7 | 42.34 |
| XA-5-15 | 0 | 75,871 | 70.5716302 | 1.942338 | 0.04468 | 0.00343 | 0.01417 | 0.00039 | 0.08782 | 0.00641 | 0.1 | 104.97 | 90.7 | 2.51 | 85.5 | 5.98 |
| XA-5-16 | 0 | 151,920 | 76.5110941 | 1.873263 | 0.03887 | 0.00258 | 0.01307 | 0.00032 | 0.0705 | 0.00446 | 0.1 | 0 | 83.7 | 2.04 | 69.2 | 4.23 |

## 6. Discussion

### 6.1. Timing of Sn Mineralization

Cassiterite is the main ore mineral in tin polymetallic deposits; thus, directly dating cassiterite can provide precise constraints on the timing of mineralization processes. With recent improvements in analytical techniques, the LA-ICP-MS U-Pb dating of cassiterite ($SnO_2$) has proven to be a powerful tool for dating tin polymetallic deposits [7,12,13,15–18,72]. Li et al. [10] discussed the reliability of using LA-ICP-MS to determine the U-Pb age of cassiterite and proposed the relative merits and reliability of isochron ages, concordia ages, and Tera-Wasserburg lower intercept ages.

In this study, we applied the LA-ICP-MS technique to obtain the U-Pb chronology of cassiterites from the two types of ores (altered rock- and tourmaline vein-type ores) in the Xi'ao Cu-Sn polymetallic deposit. The $^{207}Pb/^{206}Pb$-$^{238}U/^{206}Pb$ Tera-Wasserburg concordia lower intercept ages of 83.3 ± 2.1 Ma to 84.9 ± 1.7 Ma (Figure 7) obtained from our data agree well with those of the Kafang (one of the five deposits in Gejiu district) cassiterite (84.4 ± 2.0 Ma) obtained by Guo et al. [35], and those of the Gaosong (one of the five deposits in Gejiu district) cassiterites (83.5 ± 2.1 Ma to 85.1 ± 1.0 Ma) obtained by Guo et al. [14]. The U-Pb system in cassiterite has a high closure temperature, and cassiterite has stable chemical properties and is resistant to hydrothermal alteration. Therefore, we conclude that the timing of Sn mineralization in the Xi'ao deposit occurred at 83.3 ± 2.1 Ma to 84.9 ± 1.7 Ma.

Numerous geochronological studies have been performed on the Laoka granite. The reported U-Pb age for the Laoka equigranular biotite granite is 85.0 ± 0.85 Ma [41,43]. Furthermore, the U-Pb ages (83.3 to 84.9 Ma) of the cassiterites obtained in our study are consistent with the previously mentioned ages, which indicates that the altered rock- and tourmaline vein-type ores in the Xi'ao Cu-Sn polymetallic deposit have a close temporal relationship with the Late Cretaceous granitic magmatism.

### 6.2. Color, CL Images and Raman Spectrum

The color variation of cassiterite has been attributed to elemental isomorphous substitution in the mineral lattice [7,73,74], or Fe content [75,76] or $Fe^{2+}/Fe^{3+}$ ratio [77]. However, in our cassiterites collected from Xi'ao deposit, there were no obvious relationships between Fe + Nb + Ta + Ti, Fe, or W contents and the color variation (Figure 8). Moreover, previous studies also show W and U contents are the main factors controlling the color variation of Gaosong cassiterites [14,78,79]. Both of the results illustrate other mechanisms may exist, which still need further study.

Some studies show Fe, Ti, and W contents are related to the luminescence intensity [80–84]. Figure 5 shows that the O units had elevated Ti content, and the H units usually had no or little Ti. The W content had no distinct relation with luminescence intensity. Therefore, the Ti content played an important role in the luminescence intensity of CL images of the Xi'ao cassiterites. In addition, $A_{1g}$ was one of the characteristic peaks of cassiterite, whose Raman frequency could be significantly affected by impurities [69–71,85]. In Figure 6, low-frequency drift of $A_{1g}$ may have been caused by the substitution of Sn by Nb, Ta, Fe, and Mn [70].

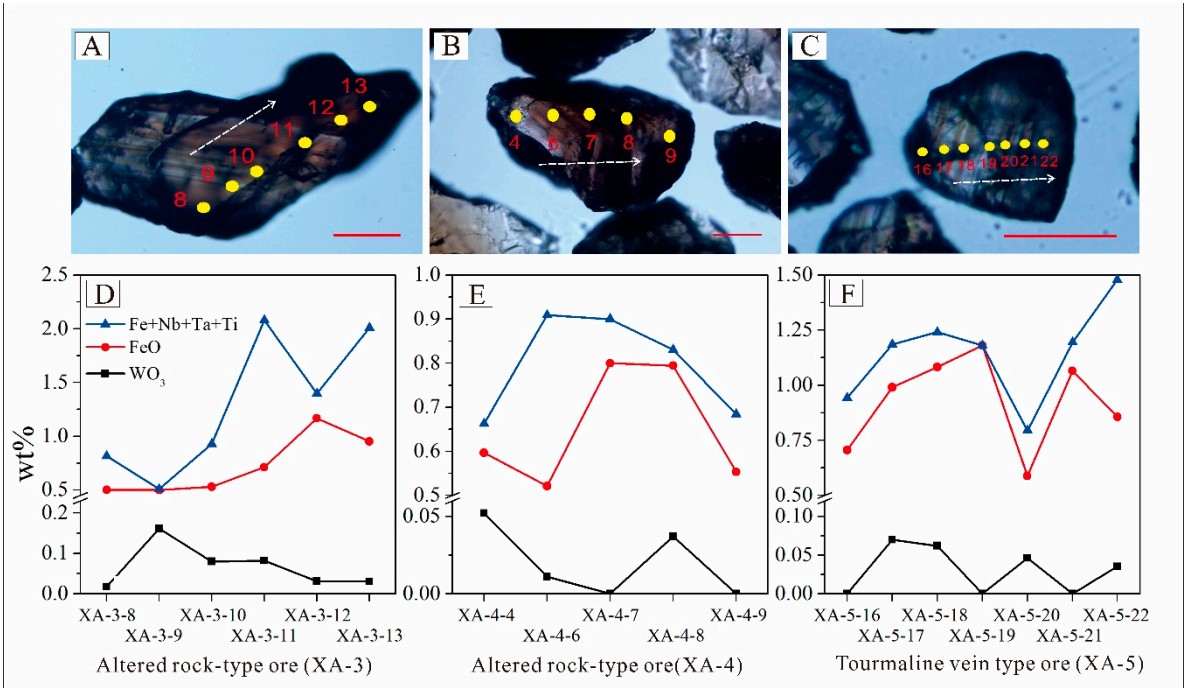

**Figure 8.** Relations between color and WO₃, Fe, and Fe + Nb + Ta + Ti contents in different types of cassiterite samples. A and D are XA-3; B and E are XA-4; C and F are XA-5.

### 6.3. Metallogenic Conditions

The $\delta^{18}O$ values of cassiterite samples were indistinguishable and exhibited a narrow range from 3.9‰ to 4.7‰. The $\delta^{18}O_{H2O}$ values were calculated using the formula based on the $\delta^{18}O$ values of cassiterite that may represent the $\delta^{18}O$ values of ore-forming fluid, which were in the range from 7.16‰ to 8.25‰. The oxygen isotope data are consistent with the values of magmatic water ($\delta^{18}O_{H2O}$ = 5.5‰ to 9.5‰) proposed by Ohmoto [86] and Sheppard [87]. Additionally, previous studies of rocks and ore minerals from the Gejiu district have reported oxygen isotopic compositions (Table 7): the whole-rock $\delta^{18}O$ of Laoka granite was 11.85‰ [88], and the calculated $\delta^{18}O_{H2O}$ of Laoka granite was 9.3‰ [57], the $\delta^{18}O_{biotite}$ of Laoka granite was 8.58‰ [88]; the $\delta^{18}O$ values of Gejiu Formation carbonate rock limestone, marble, and dolomite were 27.31‰, 22.25‰, and 18.70‰, respectively [89]. The $\delta^{18}O_{H2O}$ values of cassiterite samples are consistent with the calculated $\delta^{18}O_{H2O}$ of Laoka granite rather than those of carbonate, and the H-O isotopic data of quartz in different mineralization stages indicate that ore-forming fluids were mostly derived from magma as well as the late-stage addition of meteoric water [50,88]. Therefore, we conclude that the ore-forming fluids were derived from magmatic hydrothermal system.

**Table 7.** The $\delta^{18}O$ of other geological samples in Xi'ao Cu-Sn polymetallic deposit.

| Samples | Minerals | $\delta^{18}O_{mineral}$ (‰) | Reference |
|---|---|---|---|
| | Whole rock (4) | 11.85 | [88] |
| Laoka granite | Biotite (2) | 8.58 | |
| | Quartz | 12.40 | [57] |
| Ore-bearing quartz vein | Quartz | 12.60 | [88] |
| Gejiu Formation limestone | Limestone | 27.31 | |
| Gejiu Formation marble | Marble (2) | 22.25 | [89] |
| Gejiu Formation dolomite | Dolomite | 18.70 | |
| Basalt amygdala | calcite | 19.29 | |

Numerous studies have analyzed the geochemistry of cassiterites from different metallogenic environments and have observed systematic variations in their chemical compositions [3,90,91]. Cassiterites from the Xi'ao deposit had high Fe and Mn contents, which plot in the field of hydrothermal cassiterites (Figure 9). The Fe and Mn contents of cassiterites from the Xi'ao deposit were obviously higher than those of cassiterites from the Gaosong deposit. The (Ta + Nb)/(Fe + Mn) atomic ratio of cassiterite samples varied greatly, and all data fell above the line of (Ta + Nb)/(Fe + Mn) = 1 (Figure 9). This result could be attributed to the excess Fe present in most cassiterite samples (Table 3). The (Ta + Nb)/(Fe + Mn) atomic ratios and their correlations demonstrate that various charge compensation mechanisms occur when $Sn^{4+}$ is substituted [4,90]. The cations could be incorporated into the cassiterite structure according to the equation $3Sn^{4+} = (Fe, Mn)^{2+} + 2(Nb, Ta)^{5+}$ and $2Sn^{4+} = Fe^{3+} + (Nb, Ta)^{5+}$ [74,90]. The remaining Fe would then be incorporated into the crystal by the substitutions $Sn^{4+} = Fe^{3+} + H^+$, $Sn^{4+} + O^{2-} = Fe^{3+} + OH^-$ [90], and Ti would be incorporated by the mechanism of $Sn^{4+} = Ti^{4+}$.

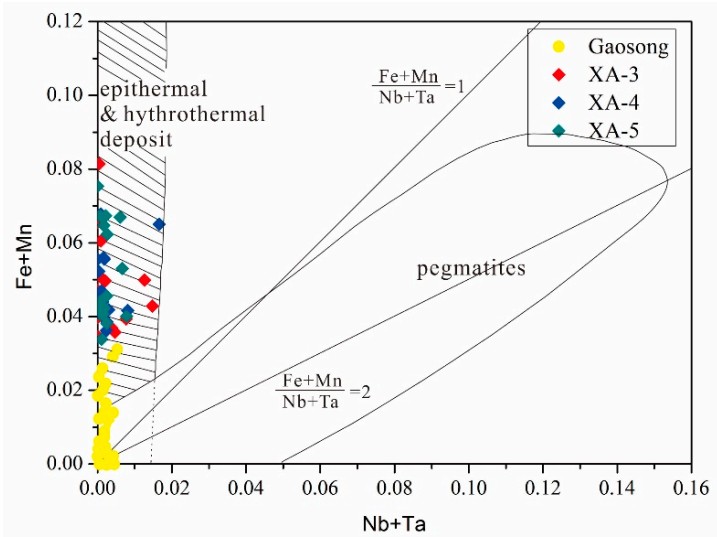

**Figure 9.** Covariation plot of Nb + Ta with Fe + Mn in cassiterite grains from the Xi'ao Cu-Sn deposit (EPMA data) (modified from [14]).

The trace element concentrations of cassiterites can provide information about the ore-forming fluid. For example, the high field strength elements (HFSEs, Nb, Ta, and Ti) contents of the cassiterites related to magmatic hydrothermal fluids are elevated when compared to those of cassiterites associated with sedimentary or metamorphic hydrothermal fluids [2,6,7,92,93]; for example, studies have shown that in the Furong tin deposit, HFSE concentrations of cassiterite in granite are high (Sc: 2.81~22.9 ppm; Ti: 81.1~1952 ppm; Nb: 8.94~1046 ppm; and Ta: 0.063~74.0 ppm). In contrast, cassiterite in carbonate rock, which is greatly influenced by country rock, has less Sc (0.181~1.80 ppm), Ti (3.44~75.2 ppm), Nb (0.002~0.129 ppm), and Ta (0.001~0.022 ppm) [7]. Cassiterite collected from a magmatic hydrothermal tin deposit related to granite in northern Portugal contains high contents of $Nb_2O_5$ (0.05 wt %~1.12 wt %), $Ta_2O_5$ (0.23 wt %~2.37 wt %), and $TiO_2$ (0.24 wt %~0.45 wt %) [6]. Cassiterite collected from the Yunlong metamorphic hydrothermal tin deposit contains low contents of HFSEs, e.g., Nb (0.17 ppm ~0.31 ppm), Ta (0.07 ppm ~0.09 ppm), Sc (2.34 ppm ~2.48 ppm), and Ti (115~128 ppm) [2]. All of

our studied samples contained relatively higher contents of $Nb_2O_5$ (0~0.592 wt %), $Ta_2O_5$ (0~0.376 wt %), and $TiO_2$ (0~1.448 wt %) (Table 3), indicating that the sources of the ore-forming fluid are enriched in HFSEs. The contents of HFSEs were high in the Laoka granite (Nb: 53.86ppm, Ta: 19.4 ppm; Ti: 179.85 ppm [94]) but very low in the Gejiu Formation carbonate rocks, both of which are related to the Laochang tin polymetallic deposit [57,94]. Hence, the high contents of HFSEs in cassiterites may be derived from the Laoka granite, and these cassiterites likely formed in a metallogenic environment that was largely affected by granitic magma. In addition, as shown in Table 8, the S isotopic compositions of different sulfide minerals (pyrrhotite, arsenopyrite, chalcopyrite) indicate that their $\delta^{34}S$ values (−0.2‰~2.9‰, [38,94]) fall within the range of the $\delta^{34}S$ values for granite (−3.7‰ to 0.1‰, [23]), which is obviously lower than that of the Gejiu Formation carbonate rock (7.14‰ to 11.1‰, [57]), and interpreted to be of magmatic origin. All of the above data demonstrate that the Laoka granite provided the source material for tin mineralization.

**Table 8.** Composition of sulfur isotope in the Xi'ao Cu-Sn polymetallic deposit.

| Minerals | $\delta^{34}S_{V\text{-}CDT}$ (‰) | Reference |
|---|---|---|
| Laoka granite | −3.7~0.1 | [23] |
| Carbonate | 7.4~11.1 | [57] |
| Pyrrhotite | −0.2 | [94] |
| Chalcopyrite | 1.9 | |
| Pyrrhotite | 0.2 | [38] |
| Arsenopyrite | 2.2 | |
| Chalcopyrite | 2.9 | |

As important carriers of Sn, W, and other metal elements, F and Cl play a key role in the separation, migration, and enrichment of Sn, W, and HFSEs (e.g., Nb, Ta, Ti) from melt. Studies have shown that the solubilities of Nb and Ta significantly improve in fluoride solutions [95,96], as does that of Ti; these solubilities are 20–200 times higher in fluorine solutions than in pure water [97]. Within a reducing environment, the solubility of Sn in fluid can obviously increase due to the fact that $Sn^{2+}$ and $Cl^-$ can form more stable complexes [98,99]. The Laoka granite is peraluminous, highly evolved, and fractionated, and has a high fluorine content of 2500 ppm and a high F/Cl value (8.26) [41–43,56]. The abundance of fluorite associated with these ores also supports the interpretation that the ore-forming fluid contained a considerable amount of fluorine. Hence, fluorite mineralization is attributed to low pH and copious availability of $Ca^{2+}$ derived from dissolution of the host limestone [100]. Under these conditions, it can be inferred that $F^-$ beside $Cl^-$ played possible roles as relevant complexing agents of Sn. Previous studies have shown that, during magmatic differentiation, fluorine preferentially enters the melt until it becomes saturated in a liquid phase due to its highly incompatible behavior [50,101]. Meanwhile, fluorine may also decrease the viscosity, density, and solidus temperature of magma, thus extending the length of fractional crystallization and facilitating the extraction of tin, copper, and other ore-forming elements in the residual liquid phase [101–103]. The value of $D_F$ fluid/melt increases exponentially with increasing F content in the melt [104]; thus, the F content of the exsolved fluid from the F-rich melt should be high. During the late stage evolution of a F-rich magma, the ore-forming fluid formed, which was enriched in metallogenic elements (e.g., Sn, W) and HFSEs (e.g., Nb, Ta, Ti). As the ore-forming fluid migrated upward, it interacted with the Gejiu Formation carbonate rocks and was accompanied by the addition of meteoric water during the late metallogenic stage—this resulted in the formation of cassiterites with high contents of HFSEs in different regions.

### 6.4. Ore Genesis of the Xi'ao Deposit

At present, most large and superlarge tin deposits are of magmatic hydrothermal origin and are closely temporally and spatially associated with granites [105–107]. However, tin deposits that record other geneses, such as the Portugal Neves Corvo VMS Cu-Sn deposit [108,109] and the China Yunnan Yunlong metamorphic hydrothermal tin deposit have also been found [2]. To the best of our knowledge, deposits of types similar to the Xi'ao tin deposits described here have not been reported

previously. This deposit allows us to study cassiterite mineralization more comprehensively. In this study, the colors, Raman spectra, elemental compositions, oxygen isotope results, and U-Pb geochronology indicate that the cassiterites from the Xi'ao deposit formed in the Late Cretaceous and are genetically related to Laoka biotite granite.

The ore bodies of the Xi'ao Cu-Sn polymetallic deposit are hosted within the granite, which extends to 400 m. In addition, the fluorite and the potassic alterations in the ore bodies suggest that the fluid system is rich in potassium and fluorine. The sulfur and oxygen isotopes characteristic of the sulfides and cassiterite indicate a magmatic origin [38,57,88,94]. Thus, the Xi'ao deposit may be classified as a magmatic hydrothermal deposit. Moreover, the younger U-Pb ages for the cassiterites reported here compared to the age of the Laoka granite corroborates this interpretation. The occurrence and alteration of the ore bodies provide useful information for exploring the altered rock-type Cu-Sn polymetallic deposit in Xi'ao.

## 7. Conclusion

The U-Pb Tera-Wasserburg concordia lower intercept ages of two samples of altered rock-type ore and one sample of tourmaline vein-type ore in the Xi'ao Cu-Sn polymetallic deposit are $83.3 \pm 2.1$ Ma, $84.9 \pm 1.7$ Ma, and $84.0 \pm 5.6$ Ma, respectively. These ages are highly consistent with the U-Pb age of Laoka granite, which indicates that mineralization has a close temporal relationship with the Late Cretaceous granitic magmatism. The peak values of $A_{1g}$ were shifted to a lower frequency, possibly due to the substitution of Sn by Nb, Ta, Fe, and Mn. The $\delta^{18}O$ values of cassiterite samples and the $\delta^{18}O_{H2O}$ values of ore-forming fluid indicate that ore-forming fluids were mostly derived from magma. The high Fe and Mn contents show that the cassiterites belong to hydrothermal cassiterites. The Nb, Ta, and Ti contents indicate that cassiterites in the Xi'ao deposit likely formed in a metallogenic environment that was largely affected by granitic magma. The later-stage hydrothermal activity dominated by Cl- and F-rich fluids was responsible for cassiterite deposition. Thus, we conclude that the Xi'ao deposit is a magmatic hydrothermal deposit.

**Author Contributions:** Conceptualization, Y.Z. and S.C.; Investigation, Y.Z., S.C., Y.H., J.Z., X.T. and X.C.; Data curation, Y.Z.; Formal analysis, Y.Z.; Funding acquisition, S.C.; Project administration, S.C; Writing—original draft preparation, Y.Z.; Writing—review and editing, Y.Z. and S.C.

**Funding:** This work was financially supported by the National Key R&D Program of China (2016YFC0600509); a project funded by China Geological Survey (DD2016005232); and the National Natural Science Foundation of China (91755208).

**Acknowledgments:** We thank Shuiyuan Yang and Kuidong Zhao in Geological Processes and Mineral Resources (GPMA) for assistance with data processing and interpretation. We want to thank Huan Tian for improving the presentation of the early version of the manuscript. We are also grateful for the reviewer's constructive comments and suggestions.

**Conflicts of Interest:** The authors declare no conflict of interest. No potential conflict of interest was reported by the authors.

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
