# Peer review of "U-Pb Ages, O Isotope Compositions, Raman Spectrum, and Geochemistry of Cassiterites from the Xi’ao Copper-Tin Polymetallic Deposit in Gejiu District, Yunnan Province"

_minerals, doi:10.3390/min9040212_

Round 1

Reviewer 1 Report

The paper provide an important set of new data for an interesting ore deposit; however, I suggest a wide revision of the first part, which lacks in particular: 1) of a minimum petrographic/petrological/metallogenic description of the granite involved into the mineralizing processes, and, more important 2) of a mineralogical/petrographic characterization of the studied mineralization, in particular of the "altered granite- type" (a variety of greisen?). Consequently, the use of the analytical data in discussion chapter is heavily penalized, in particular for what concerns the metallogenic implications.

See  the attached file for my comments.

Author Response

Dear reviewer,

Thank you for your comments concerning our manuscript entitled “U-Pb ages, O isotope, Raman spectrum and geochemistry of cassiterites from the Xi’ao copper-tin polymetallic deposit in Gejiu district, Yunnan Province” (ID: minerals-461444).

We have studied comments carefully and have made correction which we hope meet with approval. The main corrections in the paper and the responds to your comments are described in the word.

Sincerely.

Shouyu Chen

Reviewer 2 Report

The paper " U-Pb ages, O isotope, Raman spectrum and geochemistry of cassiterites from the Xi’ao copper-tin polymetallic deposit in Gejiu district, Yunnan Province" by Yuehua Zhao and co-workers presents cathodoluminescence (CL) images, Raman spectra, electron microprobe analyses, oxygen isotope analyses and LA-ICP-MS U-Pb analyses of cassiterite samples from altered rock-type and tourmaline vein-type ores from the  Xi’ao Cu-Sn polymetallic deposit.

The paper is interesting and suitable for publishing in Minerals. However, the overall style is poor and many paragraphs are not clearly written needing rephrasing. The manuscript must be checked by a native English speaker

Scientific soundness is right, however, in lines 358-359 (section 6.3), the authors should compare with the fluids that could be in equilibrium with the discussed materials. 

The section on Raman spectra and color of cassiterite could be substantially shortened as this topic is not related with the main objectives of the paper.

References are needed in some parts of the text (see annotated pdf). This point is important to avoid plagiarism problems.

Overall, minor revision is necessary, however important rewriting is necessary before its acceptance.

See annotate pdf for additional comments.

Best regards 

Author Response

(The authors gave the same response as above.)

Round 2

Reviewer 1 Report

I greatly appreciated the effort made by the authors to meet the previous observations. In particular, the work has been integrated in the parts of the geology and petrology of granitoids involved in the mineralizing processes, as well as in the petrographic and mineralogical description of the analyzed samples. As a result, the discussion appears more mature and more supported by field evidences and analytical data. In my opinion, the work fully deserves to be published.

Author Response

Dear reviewer,

Thank you for your comments concerning our manuscript entitled “U-Pb ages, O isotope, Raman spectrum and geochemistry of cassiterites from the Xi’ao copper-tin polymetallic deposit in Gejiu district, Yunnan Province” (ID: minerals-461444).

With your help, this article can meet the requirements. Thank you very much

Sincerely.

Shouyu Chen